

# Multi-century cool and warm season rainfall reconstructions for Australia's major climatic regions

Mandy Freund[1,2,3], Benjamin J. Henley[1,2], David J. Karoly[1,2], Kathryn J. Allen[4], Patrick J. Baker[4]

[1]School of Earth Sciences, University of Melbourne, Parkville, Victoria, 3010, Australia
[2]ARC Centre of Excellence for Climate System Science, Australia
[3]Australian-German Climate and Energy College, University of Melbourne, Parkville, 3010, Australia
[4]School of Ecosystem and Forest Sciences, University of Melbourne, Richmond, Victoria, 3121, Australia

*Correspondence to*: Mandy Freund (mfreund@student.unimelb.edu.au)

**Abstract.** Australian seasonal rainfall is strongly influenced by large-scale ocean-atmosphere climate influences. In this study, we exploit the links between these large-scale precipitation influences, regional rainfall variations, and palaeoclimate proxies in the region to reconstruct Australian regional rainfall between four and eight centuries into the past. We use an extensive network of palaeoclimate records from the Southern Hemisphere to reconstruct cool (Apr-Sep) and warm (Oct-Mar) season rainfall in eight natural resource management (NRM) regions spanning the Australian continent. Our sub-annual

rainfall reconstruction aligns well with independent early documentary sources and existing reconstructions. Critically, this reconstruction allows us, for the first time, to place recent observations at a sub-annual temporal resolution into a pre-instrumental context, across the entire continent of Australia. We find that recent 30-year and 50-year trends towards wetter conditions in tropical northern Australia are highly unusual in the multi-century context of our reconstruction. Recent cool season drying trends in parts of southern Australia are also very unusual, although not unprecedented, across the multi-

century context. We also use our reconstruction to investigate the spatial and temporal extent of historical drought events. Our reconstruction reveals that the spatial extent and duration of the Millennium drought (1997-2009) appears either very much below average or unprecedented in southern Australia over at least the last 400 years. Our reconstruction identifies a number of severe droughts over the past several centuries that vary widely in their spatial footprint, highlighting the high degree of diversity in historical droughts across the Australian continent. We document distinct characteristics of major

droughts in terms of their spatial extent, duration, intensity, and seasonality. Compared to the three largest droughts in the instrumental period (Federation drought [1895-1903], World War II drought [1939-1945], and the Millennium drought [1997-2005]), we find that the historically documented Settlement drought [1790-1793], Sturt drought [1809-1830] and the Goyder Line drought [1861-1866] actually had more regionalised patterns and reduced spatial extents. This seasonal rainfall reconstruction provides a new opportunity to understand Australian rainfall variability, by contextualising severe droughts

and recent trends in Australia.



## 1 Introduction

Australia's climate varies between extreme states of severe dry conditions and devastating wet episodes affecting large areas of the continent. Shaped by high variability and persistence, floods, heat waves and droughts, Australia is highly vulnerable to changes in the climate system. One reason for the diversity in climate states is the influence of, and interactions among, large-scale ocean-atmosphere modes of variability. These include the El Niño Southern Oscillation (ENSO), the Indian Ocean Dipole (IOD), the southern annular mode (SAM), and atmospheric characteristics such as the strength and location of the Subtropical Ridge (STR) and the presence of atmospheric blocking (BLK). Critically, these tropical and extra-tropical modes of variability operate at and across different temporal scales and their individual and interacting influences have a strong—and diverse—seasonal and regional effects on Australia's climate (Cai et al., 2014; Drosdowsky, 1993; Larsen and Nicholls, 2009; Maher and Sherwood, 2014; McBride and Nicholls, 1983; Oliveira and Ambrizzi, 2016; Ummenhofer et al., 2011a; Wang and Hendon, 2007; Watterson, 2009; 2011).

Over the 20th century many regions in Australia have experienced prolonged pluvial and drought periods that are documented in the instrumental records. The Federation drought (1895-1903) was one of the first multi-year periods of below average rainfall since European instrumental data collection began in Australia. There were also pronounced rainfall deficits during the World War II drought (1939-1945) and the Millennium drought (1997-2005), with devastating effects on regional agriculture and the broader economy(van Dijk et al., 2013).

In addition to these discrete drought events, there have also been a number of trends observed in Australian rainfall in recent decades. While there has been a general decrease in rainfall, particularly across southern Australia, these changes appear to have strong seasonal and regional components. For example, rainfall has declined in autumn across southern Australia (Larsen and Nicholls, 2009; McBride and Nicholls, 1983; Murphy and Timbal, 2008; Timbal et al., 2006), in the south-west during winter (Allan and Haylock, 1993; Cai and Cowan, 2008; Hope et al., 2009) and in southeast Queensland during summer (Smith, 2004; Speer et al., 2009). At the same time, regions in the north have received increasing rainfall (Feng et al., 2013; Taschetto and England, 2009; Wardle, 2004). The Millennium drought, observed most severely in south-western and south-eastern Australia, was predominately due to deficits in cool season rainfall (Verdon-Kidd and Kiem, 2009a).

Given the presence of decadal (or longer) variability in the known climate drivers, short observational records are unlikely to provide a reliable estimate of the full extent of natural variability in Australia's climate system. In building a picture of the future likelihood of observed late 20th Century trends continuing and the underlying likelihood of prolonged drought, it is essential that we understand the longer-term climatic context and its sources of variability. Palaeoclimate data can provide a unique window into long-term rainfall variability and emerging spatial and temporal trends. Such knowledge has practical applications for water resources management, seasonal forecasting, and future climate predictions.





There have been a number of palaeoclimate reconstructions of hydrological variables in Australia (Allen et al., 2015)(Cullen & Grierson 2008; Gallant & Gergis 2011; Gergis et al. 2012; Lough, Lewis & Cantin 2015; Heinrich et al.,2009). Palmer et al. (2015) recently introduced the Australia and New Zealand Drought Atlas (ANZDA), using the approach developed in

Asia and North America by Cook et al. (2010). The ANZDA reconstructs the past 500 years of Palmer Drought Severity Index (PDSI) for a 0.5 x 0.5 degree grid over eastern Australia and New Zealand using a network of 176 tree ring records and one coral record. Each of these reconstructions has advanced our knowledge of hydroclimatic variability at the sub-annual or annual scale for specific regions of Australia. To date, however, none have performed sub-annual reconstructions for the entire Australian continent.

The network of palaeoclimate proxies in Australia prior to the instrumental period is much sparser than for other regions such as Europe and North America. However, the strong links between large-scale remote climate drivers and Australian climate means that remote proxies can contain a useful climate signal. Several recent studies have used remote teleconnections and climate drivers to obtain skilful reconstructions (Tozer et al. 2015; Vance et al. 2015, Palmer et al.

2015). In this study, we introduce a new method to reconstruct regional rainfall by systematically relating instrumental rainfall and proxy information to remote climate influences. We utilise this process-based methodology to maximise the skill and widespread utility of our reconstructions of Australian rainfall.

Rainfall variations over the Australian continent show a large degree of spatial coherence at seasonal and longer time-steps,
due to the relatively simple terrain geometries and orography. The Climate Change in Australia report (CSIRO and Bureau of Meteorology, 2015) applied a regionalisation scheme to define eight Natural Resource Management (NRM) regions with similar climatic and biophysical features. The NRM clusters and their abbreviations are listed in Table 1 and shown on the map in Figure 1. In this study, we use a diverse network of local and remote palaeoclimate proxies to perform a reconstruction of cool and warm season rainfall in these eight NRM regions of Australia.

The aims of this study are to:

1. Consolidate relevant hydroclimate-sensitive palaeoclimate records;

2. Assess the sensitivity of the palaeoclimate records to the influences of large-scale climate influences and test the stationarity of these relationships;

3. Exploit the sensitivity of palaeoclimate proxies to large-scale climate influences and develop skilful palaeoclimate reconstructions of seasonal rainfall in eight NRM regions for several centuries into the past; and

4. Compare the occurrence of wet and dry periods in the past to those in the instrumental period to provide a longer-term context for recent observed events and trends.



Our study is organised as follows: Sections 2 and 3 describe the data and our methods, respectively. Section 4.1 presents a summary of the regional rainfall signature of modes of variability in the instrumental period. Section 4.2 presents the results of the reconstruction. In Section 4.3, we present an investigation of the trends, droughts and extreme years in a multi-centennial context, as well as a comparison to existing reconstructions. We finish by discussing these results and their

broader implications in Section 5.

## 2. Data

### 2.2 Instrumental data

Our analysis is based on the Australian Bureau of Meteorology's gridded monthly precipitation dataset from the Australian Water Availability Project (AWAP) (Jones et al., 2009). Seasonal and regional averages are computed from the gridded

observational dataset at its highest spatial resolution of 0.05°×0.05° for the period 1900-2015.

We also use several climate indices to link climate drivers with Australian rainfall (Table 1(a)). The indices describe tropical influences (El Niño Southern Oscillation (ENSO) and Indian Ocean dipole (IOD/DMI) (Saji et al., 1999)) as well as extra-tropical drivers (Southern Annular Mode (SAM), the intensity and position of the Subtropical Ridge strength (STRI and STRP (Drosdowsky, 1993)) and atmospheric blocking (BLK) (Pook and Gibson, 1999)). ENSO is described by multiple

indices, each of which relates to a different aspect of the coupled ocean-atmosphere mode. Here we use indices that are related to sea surface temperatures anomalies in the Eastern Pacific (NCT) and the Western Pacific (NWP) (Ren and Jin, 2011), the Southern Oscillation index (SOI) that measures the atmospheric component of ENSO, and the effects of Central-Pacific type events denoted by the ENSO Modoki index (EMI) (Ashok et al., 2007).

### 2.2 Palaeoclimate data

A palaeoclimate network of 185 individual records is compiled for the Southern Hemisphere (Fig. 1a). The multi-proxy network includes local and remote sites from a broad area that are related either directly or tele-connected to Australian climate (Fig. 1a). The majority of the records used are derived from the underlying network of the recently developed Australia and New Zealand summer drought atlas (ANZDA) (Palmer et al. 2015) and the Ocean2k project that is part of the PAGES (Past Global Changes) program (Tierney et al. 2015; Neukom & Gergis 2012). This network includes 131 tree-ring

records from the Australasian Pacific area, 36 coral-derived records from the tropical Pacific and Indian Oceans, and five speleothem-derived records. In addition, 13 records derived from Antarctic ice-cores are included, as they are related to relevant large-scale drivers such as SAM (Tozer et al. 2015, Vance et al. 2015). All records in the network extend back to at least 1880 CE and the majority cover the past 250 years. Approximately half the records extend back before 1600 CE. Twenty records extend back to 1200 CE or earlier (Fig. 1b). Within this network, 160 proxy records are annually resolved

records and 25 sub-annually resolved derived from corals (Fig. 1c). Sub-annually resolved records are binned into seasonal averages according to a warm (Oct–Mar) and cool season (Apr–Sep).



## 3 Methodology

### 3.1 Reconstruction

The ocean-atmosphere processes that influence Australia's hydroclimate have distinct, but variable, seasonal and geographical characteristics (Risbey et al. 2009). In this study, we first consider the relationships between the selected

climate indices and warm (Oct–Mar) and cool (Apr–Sep) season rainfall in each NRM region. The influence of each driver is determined by linear correlation for the concurrent season only. We exclude lag relationships between each driver and rainfall, which are generally weaker (Risbey et al. 2009). Relationships between precipitation and ocean-atmosphere processes can vary in strength over time (Gallant et al. 2013). We therefore used moving correlation windows (window length = 30 year) to assess statistically significant (p <0.1) correlations for temporal stability. A relationship is considered

stable if the interquartile range of windowed correlations remains of the same sign for the entire period of overlap between the two data sets. This approach ensures, on one hand, that the climate drivers have an approximately time-stable relationship with rainfall, but also allows some degree of variation in the strength of the teleconnection. The same procedure to test stability was applied to the relationships between each proxy record (as listed in Table S1) and each climate index (as listed in Table 1 (a)). Only proxies with a significant and time-stable relationship with an index were used as predictors for

NRM regions with a time-stable relationship between that same index and precipitation (Table S2).

We use a nested, composite-plus-scale (CPS) approach (Bradley & Jones 1993; Tierney et al. 2015) to reconstruct regionally averaged rainfall for each NRM region. Our CPS approach combines principal components of proxy records into regional composites based on a weighted averaging procedure. The weight, $w$, is determined by 1) the coefficient of determination

between each record and its target during the common period (1900-1984) and 2) the significance of this relationship, $w=r^2(1-p)$, where p denotes the p-values of the correlation, similar to Tierney et al. (2015). The resulting composite is re-scaled to the mean and standard deviation during the calibration period. Using a nested approach entails the calculation of multiple reconstructions, each reconstruction, or nest, extending further back in time but including fewer proxies as the proxies successively drop out (Fig. 1 b). Nests are spliced together to form a continuous reconstruction. This process

maximises the length of the final reconstruction and ensures all proxies meeting the selection criteria are used at each point in time. The common period of palaeoclimate records and instrumental data (1900-1984) is used for calibration and verification. During the common period, 60% of the data are used for calibration (equal to 52 contiguous years) and the remaining 40% (33 years) are used for verification. We assess the sensitivity of our reconstruction to different calibration and verification periods by shifting our calibration window of 52 years across the common period in steps of 5 years. These

different, but not entirely independent, calibration and verification periods are used to build an ensemble of seven reconstructions for the warm and cool seasons.





Each final regional rainfall reconstruction is evaluated against a set of skill metrics. The coefficient of determination ($R^2c$), is a measure of variance explained by the reconstruction in the calibration period, and $R^2v$ is the variance explained in the verification period. Further skill statistics include the reduction of error (RE) and the coefficient of efficiency (CE), which both indicate statistical skill by positive values (Cook & Kairiukstis 1990). Further analysis is conducted on the skillful

portion of the reconstruction (CE>0). We report our best reconstruction as that which maximises the time-integrated RE.

### 3.2 Analysis

Linear trends in the warm and cool season from instrumental and reconstructed precipitation are compared by fitting a linear trend line to the time series in 30- and 50-year moving windows. All trends are normalized and presented in histograms. All trend calculations are based on overlapping moving windows with a 1-yr time step.

We investigate multi-year instrumental and historical periods of extremely low rainfall. During the instrumental period, three major droughts are assessed. These are the Millennium (1997-2009), World War II (1935-1945) and Federation droughts (1895-1903). Since European settlement, seven historical droughts are often reported in historical and documentary records. These include the Settlement drought (1790-1793), the Murray Darling Basin drought (1797-1805), the Great Drought

(1809-1814), Sturt's drought (1809-1830), South East Australia drought (1836-1845), the Black Thursday drought (1849-1866) and the Goyder Line drought (1861-1866) (Helman, 2009).

We evaluate historical periods of drought by calculating seasonal and annual rainfall deciles based on the entire length of available data. For instrumental data, deciles are relative to the 1900-2014 long-term climatology, while reconstructed

rainfall deciles include all positively verified years. The extended reconstruction includes all verified reconstructed years up to 1984, extended for the most recent 20 years of instrumental data up to 2014. It is important to note that deciles deliver quantitative statements only in conjunction with a baseline period and duration. Resulting deciles are then categorised into: highest on record, very much above average (10), above average (8-9), average (4-7), below average (2-3), very much below average (1) and lowest on record.

Drought durations are a challenge to compare using a decile-based approach alone. We therefore apply the concept of "drought-depth-duration" (DDD), following Fiddes and Timbal, (2017) and Timbal and Fawcett, (2013) to compare droughts of different duration. We present the percentage reduction below the long-term average of dry episodes ranging from 1 to 10 years. Our seasonal rainfall reconstruction considers the long-term average of the entire reconstruction baseline

and presents the drought depth duration as the percentage reduction below this long-term average.



Ranking the seasonal rainfall totals in ascending order identifies extreme dry and wet years. The 10 highest and lowest years in the best reconstruction are reported in Table 2. As above, to provide a long-term context, the 10 driest and wettest years during the extended period (both the instrumental and the reconstruction periods) are identified.

In addition to pure statistical verification, we compare our results to other studies that have used historical documentary records and paleoclimate archives to describe or reconstruct past hydroclimatic variability. Data from other studies with sub-annual resolution are averaged into the same warm and cool seasons used as the basis for reconstructions in this study. Annually resolved data are compared to both seasons. Single location records are compared to each of our regions. For the ANZDA (Palmer et al. 2015), area averages of the NRM clusters are extracted for comparison with the NRM regions.

**4 Results**

**4.1 Regional Climate Driver Influences**

The influence of climate drivers on rainfall variability across the NRM regions is significant and pervasive. The influence of ENSO stands out across the tropical north and subtropical regions for both warm and cool seasons (Fig. 2a & b). Indeed, most low-latitude rainfall variance during the ENSO peak season (warm season) is related to ENSO (up to 44% in the Wet

Tropics). The dominant effect of ENSO decreases along a north-south gradient, with multiple-drivers becoming more important in the south. In southwest and southeast Australia (Flatlands, Southern Slopes, Murray Basin) the influence of mid-latitude pressure systems encapsulated by SAM and atmospheric blocking (BLK) increases. This north-south gradient is stronger during the cool season. In southern Australia (Southern Slopes and the Murray Basin) where cool season rainfall dominates, the strength and positive influence of the Subtropical Ridge (STRP) is important. In southeastern Australia, the

correlation between rainfall and the STRP is as strong as $r=0.78$, highlighting the importance of the subtropical ridge on rainfall. Although the influence of the STRI dominates rainfall in these regions in the cool season, both SAM, BLK and ENSO still have significant associations with mid-latitude rainfall (Fig. 2a). While conditions in the tropical Pacific have a strong influence on warm season rainfall across the continent, the conditions in the Indian Ocean (IOD) have mostly cool season impacts, except in the Wet Tropics. These results are consistent with previous studies (e.g., Risbey et al. 2009).

**4.2 The Reconstruction**

**4.2.1 Reconstruction Skill**

Our reconstruction captures 30-60% of seasonal rainfall variability across the regions. Skill statistics of the reconstruction (Fig.3) show that the variance explained during the calibration period (1934-1984) is around 37% ($R^2c$ [0.2-0.5]) for the cool

season and 34% ($R^2c$ [0.2-0.6]) for the warm season (Fig. 3a&b). During the independent verification period (1900-1933), a slightly larger magnitude of variance is explained by the reconstruction, with about 46% ($R^2v$ [0.2-0.6]) for the cool season



and 48% ($R^2v$ [0.3-0.6]) for the warm season. These high and stable proportions of variance captured by the reconstructions are found for both seasons. The RE and CE statistics are positive for all regions, indicating reconstruction skill for both seasons across Australia (Fig. 3a&b). Given that our reconstruction is based on a nested approach, with a varying set of proxies over time, we indicate on Figure 3 the timeframe during which the individual reconstructions show reliable skill for

each region and season. The years shown on each region for the $R^2$ panels (Figures 3a, b, e, and f) indicate the earliest year in which the reconstruction exceeds half of its maximum skill; years shown on the RE and CE panels (Figures 3c, d, g, and h) indicate the earliest year in which the RE and CE statistics remain positive, allowing for brief periods of negative skill, of duration less than 5 years.

During the cool season, the Central Slopes, Wet Tropics, and South and Southwestern Flatlands indicate the longest skilful (CE>0) reconstructions, extending back to 1200, 1260 and 1366, respectively. The rainfall reconstruction in the Rangelands is only skilful back to 1811 and represents the shortest skilful reconstruction during the cool season. Most of south and southeastern Australia in the cool season can be reconstructed back to at least 1749 (Southern Slopes).

In the warm season, the Southern and Southwestern Flatlands, Wet Tropics, and Murray Basin warm season reconstructions are skilful (CE > 0) for the longest period, extending back to 1200 and 1234, respectively. The warm season rainfall reconstruction in the Monsoonal North has the shortest skilful reconstruction, back to 1707. Warm season reconstructions for south and southeastern Australia indicate slightly longer periods of skill than the cool season reconstructions.

**4.2.2 Reconstruction time-series**

Seasonal time-series of our reconstructed rainfall across Australia show high rainfall variability over the instrumental period (Fig. 4) and past centuries (Fig. 5). Interannual and decadal-scale rainfall variability is well-characterised by the reconstructions. For example, the pluvial periods in the mid 1950s and late 1970s during the warm season in the eastern regions (top three panels) are remarkably well reconstructed. Although different calibration and verification periods indicate

some differences at interannual scales (grey shading), encouragingly, decadal variability is well-represented by the ensemble. In particular, seasons which dominate the total rainfall are well captured in terms of their amplitude, for example, the warm season in the Wet Tropics and cool season in the Murray Basin.

Low-frequency variability is evident in all warm and cool season reconstructions (Fig. 5 & Supp. Figs. 8, 9). At decadal time

scales, both the warm and cool seasons show synchronous decades of enhanced/reduced rainfall across different regions. For example, the very dry warm seasons (Supp. Fig. 9) observed in the 1960s (Central Slopes, Murray Basin, Rangelands, Southern Slopes, Southern and Southwestern Flatlands, Wet Tropics) are also seen in the 1760s across similar regions (Central Slopes, Murray Basin, Monsoonal North, Southern Slopes, Southern and Southwestern Flatlands, Wet Tropics).



During the 1740-50s extreme wet conditions prevailed in southern and south-eastern Australia. Rainfall during those 20 years was mostly above average for much of Eastern Australia (Central Slopes, East Coast, Murray Basin). In the 1970s similar regions saw a decade of higher than normal warm season rainfall (Central Slopes, East Coast, Rangelands, Wet Tropics). There seems to be no general pattern of prolonged decadal drought or pluvial conditions associated with specific

regions in our reconstructions.

The magnitude of the warm season pluvials during the 1970s and 1740-50s are not anomalously high based on the cool season reconstructions (Supp. Fig 8). Only East Coast and Rangelands show similarly high rainfall amounts, while other regions show average or sightly drier conditions (Central Slopes). Most of the cool season decadal trends show very distinct

regional patterns. Wetter than normal conditions in the 1870s are only evident in the Central Slopes and East Coast. Even geographically proximate regions such as the Murray Darling Basin region and the Southern Slopes show dissimilarities in terms of decadal-scale variability. Overall, warm season rainfall seems to show slightly more concurrent decades across the regions than during the cool season.

To assess the degree to which the reconstructions are seasonally distinct, shaded areas in Figure 5 highlight periods when the warm and cool season reconstructions are significantly correlated (30-year moving window, abs(correlation)>0.5). There are a small number of periods during which the cool and warm season rainfall are in phase (positively correlated, shown in red) or out-of-phase (negatively correlated, shown in blue). This can be viewed as an additional verification measure. Although some regions show periods in which warm and cool season reconstructed rainfall are in-phase, this feature is also present in

some instances in the instrumental period. For example, East Coast rainfall in the 1720s and 1760s are periods of positive correlation. During these decades, there is a degree of synchronicity, meaning that reduced/increased rainfall in the cool season is accompanied by reduced/increased rainfall in the following warm season. From 1820-1840 cool and warm season rainfall in the Southern and Southwestern Flatlands is anti-correlated, indicating opposing seasonal rainfall totals. Rainfall in the late 1970s/1980s in the Murray Basin, Southern Slopes and Wet Tropics are examples of positive inter-seasonal

correlations (Figs. 5c, f, h) in the instrumental period. Whilst some regions have short periods of synchronicity across seasons, this is observed across both the reconstruction and instrumental periods and the general pattern is one of seasonal independence.

### 4.3 Australian rainfall and drought in a multicentury context

**4.3.1 Contextualising recent rainfall trends**

Since the start of instrumental records in Australia, several major droughts and extended pluvial periods have been observed. Some influences can be regarded as temporary changes in the mean-state due to, for example, natural decadal climate





variability from the Interdecadal Pacific Oscillation (IPO, Henley et al., 2015; Henley et al., 2017). Other changes appear to be more strongly related to long-term changes in atmospheric circulation. These changes are spatially and temporally diverse, and strong interannual variability makes it hard to distinguish between low-frequency variability from externally forced long-term changes. Here we use our rainfall reconstructions to place recent observed trends in a long-term centennial

context.

The histograms in Fig. 6 summarise all 30-year and 50-year linear trends in the regional reconstructions and instrumental data over the period 1600-2014. The distributions of trends distinguish between pre-1970 variability (grey), trends since 1970 or 1950 (depending on the fitted trend length of 30/50 years; shown in light blue/red for cool/warm season) and the

trend from the most recent 30 or 50 years. The regional reconstructions show recent tendencies towards drier cool seasons in the south (Murray Basin, Southern Slopes, Southern and Southwestern Flatlands) and wetter warm seasons in the north (Monsoonal North, Rangelands, Wet Tropics). The distribution of historical trends derived from the reconstructions are of a generally Gaussian distribution. Some trends starting after 1950/1970, including the most recent trend, are shifted towards the upper and lower quartile range of the pre-1950/1970 trends. Trends starting after 1950/1970, including the most recent

trend (ending in 2014) appear unusual, but not unprecedented. In recent years, during the warm season, tropical regions in particular show a strong increase in rainfall. This is strongest in the Monsoonal North, followed by the Rangelands and Wet Tropics (Supp. Fig. 4). Some subtropical regions show a warm season decrease in rainfall, strongest in the Southern Slopes, but again not unprecedented. All regions, except the Monsoonal North, show a decline in rainfall in the cool season during the most recent 30-yr and 50-yr periods. This decline is most pronounced in the Southern Slopes, Southern and South-

Western Flatlands and the Murray Darling Basin. Cool season rainfall, which contributes the majority of subtropical rainfall, has clear negative shifts in southern Australia, compared to earlier trends. In particular, cool season rainfall in the Murray Basin saw declines over the last 30 and 50-years of the order of 90mm.

### 4.3.2 Contextualising the spatial extent and intensity of past droughts

Extended periods of low rainfall have different characteristics in their temporal and spatial structure. The assessment of drought risk depends critically on the range of estimated natural variability. Here we assess the severity of major drought episodes using deciles across both instrumental (1900-2014) and multi-century reconstruction (1600-2014) periods. Our regional rainfall reconstruction comparisons here extend the timespan of the instrumental record by a factor of four, which enables us to view droughts such as the Millennium Drought in a very long-term multi-century context. Deciles based on

different datasets, baselines and durations are shown in Fig. 7. Comparing the spatial pattern of the reconstructed droughts to the gridded and NRM region instrumental (AWAP) spatial patterns, our reconstruction depicts the intensity of drought events during the instrumental period quite well. The gridded and regional representation of the Millennium drought is indicated as the lowest on record for parts of the Southern Slopes and the Murray Basin and very much below average for the



East Coast and the South and Southwestern Australia, in agreement with other studies (Cai et al., 2014; Gergis et al., 2011; Verdon-Kidd and Kiem, 2009b). In the context of the full reconstruction period (1600-2014), the Millennium drought remains the worst drought since 1749 in the Southern Slopes (observable too in Fig. 5f) and very much below average for East Coast, Murray Basin and Southern and Southwestern Flatlands. In line with probability estimates for south-eastern

Australia by Gergis et al. 2011 and Cook et al., 2015, the 12-year period of the Millennium drought is unprecedented for the Southern Slope region. The rainfall reconstruction of the Murray Basin reveals that periods in the late 1700s and early 1800s and at the time of the Federation drought are of similar or larger reductions in rainfall over a 12-year period (Fig. 5c). The Federation drought period is also apparent in the Southern and Southwestern Flatlands reconstruction, along with other periods of rainfall reductions the late 1600s (see Fig. 5g).

The World War II drought and the Federation drought appear to be of similar character during the instrumental period in terms of their area and intensity (Fig. 7 gridded and NRM regional plots). Considering the last 400 years, the World War II drought is a period of average rainfall for all regions except the Murray Basin, Central Slopes and East Coast. In contrast, the Federation drought (1895-1903) is much higher in intensity and spatial coverage. In Fig. 7 we compare the Federation

drought during its instrumental period and full (including pre-instrumental) period. During the observational period (the latter part), the Federation drought shows only slightly below average conditions. In the multi-century context, the Federation drought shows a wider extent. Along the east coast (Central Slopes, East Coast), central parts (Rangelands, Murray Basin, Southern Slopes) and north Australia (Monsoonal North, Wet Tropics) the Federation drought has been of very much below average and lowest on record for Monsoonal North and Murray Basin.

Historical droughts during the pre-instrumental period (Table 1c, Fig. 7 and Fig S.6), as documented mainly in South Eastern Australia due to the concentration of European settlement there, are captured by the reconstruction. The Goyder line drought, Sturt's drought and the Great Drought appear to have affected only certain distinct regions. The Settlement drought shows regions clearly below average, especially coastal regions and the Murray Basin, similarly to Palmer et al., (2015) and Gergis

et al., (2010). The representation in terms of affected regions aligns very well with historical reports (e.g. historical reports by Sturt or the definition of the Goyder line). Most of the historical droughts have been below average in certain regions but none of these droughts appear to exceed the spatial extent and intensity of the three major instrumental-period droughts.

### 4.3.3 Contextualising the duration and seasonality of past droughts

We apply the concept of drought-depth-duration (DDD) to our reconstructions to further assess the duration and intensity of the different droughts (see Sec. 3.2 for details). The DDD plots provide a method to compare the temporal structure of drought periods. Additionally, our bi-seasonal reconstruction resolution provides an opportunity to investigate the seasonal nature of protracted droughts and therefore provide insight into their climatic influences and potential causes.



Using the DDD analysis we can categorise droughts into long and short-term droughts and distinguish the primary season of the droughts. The Millennium drought is among the worst droughts in terms of its duration across many of the NRM regions in southern and eastern Australia (Fig. 8, dark blue lines). In particular, during the cool season, the reduction in rainfall

extends over very long periods for central and eastern regions (Central Slopes, East Coast and Murray Basin). By comparing short periods (2<yrs) to longer periods (>6yrs) the Millennium drought is revealed as a persistent drought, with its worse impacts being felt over the longer timeframe. The Murray Darling Basin drought is predominately caused by cool seasonal rainfall deficiencies over long-periods, plus a slight accumulated rainfall deficit during the warm season in the Southern Slopes. The World War II drought had similar cool season rainfall reductions, however the warm season rainfall was also

affected. These are similar to the findings of Verdon-Kidd and Kiem (2009b). The Murray Basin, the East Coast and the Wet Tropics, all had strong reductions in both cool and warm season rainfall of similar magnitude, about 70% below the long-term mean. In some regions, the Federation Drought was a strong warm season feature (Central Slopes) while in the tropical regions (Wet Tropics&Monsoonal North), rainfall reached remarkably low values. The intensity of the Federation Drought during the first few years seems to a result of re-occurring El Niño episodes at that time and highlights its ENSO's different

effects on Australian rainfall (Ummenhofer et al., 2009). Droughts such as the Murray Basin drought can be clearly identified as an extended warm season drought, not only in the Murray Basin but also along the East Coast and the Rangelands. Periods of severe long-term rainfall reductions highlight the spatial complexity of rainfall variability. There was abnormally low rainfall during the southeastern Australia drought over an extended period of up to 10 years (Monsoonal North, Murray Basin, Southern and Southwestern Flatlands). This reduction was clearly a warm season feature that was most

severe along the East Coast and the Central Slopes. Sturt's Drought is another example of a spatially distinct localised warm season drought in South-Eastern Australia that was also expressed as a long cool season drought in the Rangelands. One of the strongest short-term drought episodes was the Black Thursday drought in which warm season rainfall was 60% below the long-term average and cool season rainfall was 30% below the long-term mean. These deficits are likely to have contributed to the severe bushfires in 1851.

### 4.3.4 Extreme years in a long-term context

Severe short-term reductions of rainfall leading to events like the Black Thursday bushfires make clear the devastating effects that single extreme seasons can have. Our seasonally resolved reconstruction provides for the first time the opportunity to assess not only extreme years, but to assess individual extreme seasons. Existing annually resolved reconstructions are likely to exhibit a specific seasonal bias, which may dilute the impacts of sub-annual effects across the

year. Seasonal windows of drought indices often exhibit an integrated signal from multiple months to possibly years (Keyantash and Dracup, 2002). The rainfall reconstructions presented in this study enable us to identify temporally finer-scale extreme seasons of above/below rainfall. We identified the driest and wettest seasons for our regions by selecting the 10 strongest events using the instrumental (1900-2014) and reconstruction (1600-2014) periods.



Extreme years identified during the instrumental period reveal regional dependencies and differentiated seasonal aspects of dry and wet years (Table 2). The record breaking wet years 2010/2011 accompanied by the strong La Niña episode were extreme during the warm season across all regions. During that time the Murray Darling Basin had the largest rainfall

anomalies since 1900 (CSIRO, 2012). In the much longer context of our rainfall reconstruction, 2010/2011 was one of the wettest warm seasons in the past several centuries, not only in Murray Basin but also in Monsoonal North, Rangelands, Southern Slopes, East Coast. (Cook et al., 2016) found similar results based on the ANZDA reconstruction of spatial variability in the PDSI. The 1979-1983 period of dry conditions in eastern Australia and the 1983 warm season and 1984 cool season wet anomalies in the East Coast, Wet Tropics and Monsoonal North are also captured in the reconstructions. The

consecutive 1927 (cool), 1928 (warm) and 1928 (cool) seasons experienced extremely dry conditions in the Central Slopes region. Interestingly, extreme cool and warm season wet conditions affected multiple regions, while extreme dry conditions appear to have occurred more often in single regions. This relationship persists into the pre-instrumental period. The known asymmetry of the different ENSO phases impacting Australian rainfall could explain those differences (Cai et al., 2012). While El Niño-induced rainfall reductions are not related to the event amplitude, the La Niña phase of ENSO produces more

extreme pluvial events. This is consistent with observations that La Niña is more strongly teleconnected to rainfall, and hence extreme rainfall events (Cai et al., 2010; King et al., 2013) than El Niño events are to rainfall deficit.

In the pre-instrumental period similar patterns of spatially widespread extreme conditions in multiple regions occur (e.g., extreme dry 1481, 1607, 1760, & 1817 (cool season), extreme wet 1759, 1826, 1871 & 1879 (cool season)). Conditions

affecting multiple regions with similar magnitudes were rarer for wet than dry extremes. Warm season rainfall during the first half of the 18th century (1720, 1731, 1732, 1740, 1752) was wettest across multiple regions, while dry conditions of similar magnitude occurred most frequently during the latter half of the 18th century (1761, 1760, 1814) affecting the East Coast, Southern Slopes and the South-Western Flatlands. The intensity of widespread extreme conditions such as 2010/2011 (wet) or 1982/1983 (dry) is much reduced during the pre-instrumental period. Based on our reconstructions pre-instrumental

seasons with an amplitude comparable to the 2010 pluvial include 1759 (East Coast) or 1826 (Central Slopes) and were only extreme across a few regions. Seasonally explicit extremes highlight the value of the finer temporal resolution resolved by these reconstructions. In 1833, East Coast reconstructed rainfall shows extreme dry conditions during the cool season followed by extremely wet conditions in the following warm season.

**4.6 Comparing our reconstruction with previously published**

Here we compare our results with published studies based on palaeo-records and early documentary compilations. We begin by comparing our results to the first spatially resolved reconstruction of Australian and New Zealand summer drought variability, the ANZDA (Palmer et al., 2015). As there is significant overlap in proxy records used in this study and in Palmer et al (2015), the two studies are not independent in their source data. Nevertheless, ANZDA is based on tree-ring



records and a single coral record (that is not seasonally resolved), whereas our reconstruction includes seasonally resolved corals, as well as speolethem and ice-core records. In addition, the ANZDA drought reconstruction differs substantially in its temporal and spatial resolution, since it is a point-by-point gridded reconstruction, and targets PDSI during the warm season. Figure 9 shows the correlations between the warm season PDSI reconstruction (ANZDA) with our cool and warm season

rainfall reconstruction. Non-significant correlations during the cool season and highly significant warm season correlations, of up to 0.67 for the East Coast region, highlight agreement between our seasonal reconstruction and the summer season hydroclimatic features detected by the PDSI reconstruction. It also reiterates the highly seasonal nature of ANZDA and its bias (intentional) towards only warm season drought compared with our reconstructions. The strong temperature dependence of PDSI may explain some of the differences inland and why large parts of central Australia and all of Western Australia

were not resolved by the ANZDA reconstruction.

The study by Ashcroft et al. (2014) provides detailed early documentary records of fine temporal resolution across southeastern Australia that are entirely independent of the data used in this study. The temporal coverage of the documentary records, however, is often less than 30 years. Nevertheless, there are strong positive correlations between the documentary-

based record and the rainfall reconstruction for both the warm and cool seasons over the period 1832-1859 (Fig. 9). There are positive correlations between these records and our rainfall reconstructions across large areas in southeastern Australia, especially the Murray Basin (cool season). Figure 9 also shows a comparison of our reconstructions with rainfall variability as recorded in 18[th]- 19[th] century (1788 – 1860) records from the populated coastal centres (Ashcroft et al., 2014; Fenby and Gergis, 2012). Years classified by Fenby and Gergis (2012) as dry or wet conditions are partially reflected in our Central

Slopes reconstruction. The years 1790-93, 1810-13, and 1836-37 are consistently classified as dry years, whereas 1788, 1806 and 1830 coincide with years classified as wet. Consecutive years of dry conditions from 1820-28 appear to coincide, but single years such as 1801 and 1802 appear to be not in agreement with our reconstruction. This may be the result of more localised rainfall variability than is resolved by our regional Central Slopes reconstruction. Discrepancies between our reconstruction and documentary sources might also arise from the use of annual means, which might dilute a seasonal signal.

Comparing annual means of our seasonal reconstructions (not shown) extreme years such as 1860 (East Coast) stand out and mostly agree with the wet and dry classified years found by Gergis & Ashcroft (2012) for South East Australia.

## 5 Discussion and Conclusions

In this study we used an extensive palaeo-climate proxy network derived from tree-rings, corals, ice cores, and speleothem records to reconstruct precipitation in the eight NRM regions across Australia. This is the first Australia-wide reconstruction

of seasonal rainfall extending ~400 years into the past. The relationships between climate process indices were evaluated for the strength and stability of their relationship with precipitation in each NRM region for both the warm and cool seasons. We simultaneously assessed the strength and stability of relationships between individual proxy records and these same





processes. This process-based approach enabled the reconstruction of precipitation based on teleconnections with major processes known to be related to Australian rainfall (Hendon et al., 2007; Risbey et al., 2009). The screening of predictors based on the strength and stability of their relationship with the climate indices constrains the inclusion of predictors during the instrumental period but relies still on a continuing stationarity assumption on multidecadal timescales. All

reconstructions successfully verified over the 1749-1984 period, and many back to the early 16[th] century. A comparison with the ANZDA reconstruction showed a high-level of agreement with Eastern Australian drought conditions during the warm season. Independent high-resolution early documentary records by Ashcroft et al. 2014 compared well with the cool and warm season reconstructions. We also assessed extreme years of high and low rainfall with published documentary sources. A majority of years previously identified as having anomalously high or low rainfall events agree well with our seasonal

reconstructions. Most of those comparisons are confined to southeastern Australia due to regional biases in documentary records. Nevertheless, these additional verification approaches highlight the quality of our seasonal reconstruction and its ability to represent past rainfall variability.

On decadal to multi-decadal time scales, substantial low-frequency variability is present across the regions and seasons. An

investigation of recent trends revealed evidence for unusual tendencies towards wetter conditions in the north in the warm season and drier conditions in the south in the cool season. Northern regions (Monsoonal North, Wet Tropics) have experienced an increase over the last 30-50 years in rainfall, predominantly during the warm (wet) season (Nicholls, 2006; Taschetto and England, 2009) when the majority of rainfall occurs. The significance of this increase is difficult to determine due to high intrinsic variability in the tropical North. The extended baseline of our reconstruction helps to place these

changes into a long-term context. In particular, after the 1970s the increase of warm season rainfall in the Monsoonal North (and the Rangelands) is highly unusual relative to past centuries. Possible mechanisms could be strengthening of the monsoon modulated by the ENSO conditions in the western Pacific (Fig. 2), intensification and shifts in deep convection related to the monsoon trough (Taschetto and England, 2009), and anthropogenic forcing enhancing rainfall and cloud formation (Cai et al., 2014). Possible enhancement of atmospheric pressure systems over the Indian Ocean in conjunction

with a strengthening of SAM (Feng et al., 2013) is a further possibility. From the correlation analysis of possible drivers (Fig. 2) neither the IOD nor SAM are significantly correlated with rainfall in the northern regions, so this latter possibility seems unlikely.

The tendency towards drier conditions in southern Australia is less clear. Our analysis showed that the most recent decline in

rainfall in the Southern Slopes is within the range of variability given by the reconstruction. The cool season decline in the south is often associated with the intensification of the subtropical ridge (Timbal and Drosdowsky, 2012; Timbal et al., 2006), changes in large-scale atmospheric circulation such as the Indian Ocean (Ummenhofer et al., 2011b), and the observed upward trend in SAM (Cai and Cowan, 2006). All of these processes are significantly linked to interannual rainfall variability in the south (Fig. 2), but are not particularly unusual in terms natural variability resolved by the reconstruction.





Strong decadal to multidecadal variability can be observed across all of the regions that account for declines similar to the most recently observed trends. At least in terms of 30- to 50-year trend periods, the declining trends are within the range of intrinsic reconstructed variability. The declining trend of rainfall during the cool season is particularly strong in the Murray Basin. The most recent trends starting after 1950s and 1970s are not unprecedented, but are below the 25$^{th}$ percentile,

pointing towards a drying tendency. Further work could specifically focus on the long-term declining trend for the cool season in South and Southwestern Australia (Fig. 6) observed since around 1910.

In order to assess the severity of droughts in a long-term context, reconstructions need to be consistent with the representation of droughts in the instrumental record. All three major droughts during the instrumental periods are well

represented in terms of their spatial extent, intensity and duration (Fig. 7) (cf. Verdon-Kidd and Kiem, 2009b). The three protracted droughts remain severe when considered in the ~400-year context provided by our reconstructions, especially in south and southeastern Australia. Although all three droughts have very distinct spatial footprints, the spatial extent and concurrent drought conditions across broad areas are quite unique compared to historical droughts. The Millennium drought stands out as an unprecedented drought in the Southern Slope region (Fig. 7). The World War II drought seems not to be as

exceptional when considering the full reconstruction period back to 1600; only central and eastern regions were very much below average. The Federation drought, which was mostly prior to the observational period, is confirmed as one of the worst periods of suppressed rainfall in the Murray Basin and the Monsoonal North region. Compared to droughts during the longer period, these three major droughts affected large parts of Australia, whereas pre-instrumental drought episodes such as the Great Drought appeared to be much more regionally constrained.

The spatial extent of various droughts and pluvial episodes may help to identify the specific drivers behind the individual events. For 20$^{th}$ Century droughts, the interaction of climate modes and anthropogenic warming may have played a significant role in the drought episodes (Cai et al., 2014). However, the longer historical perspective provided by our rainfall reconstructions could help to attribute those factors in further studies by considering individual climate modes and their

interactions.

New insights about drought characteristics are derived from the comparison of instrumental and historical droughts of different duration. Different contributing processes are difficult to examine with only a few events during the instrumental period. Droughts confined to a specific season can be distinct and classified by intensity, extent and duration. Again, the

three major drought events during the instrumental period stand out in terms of their intensity and number of regions affected. The Millennium drought as a cool season feature and the Federation drought as a warm season feature clearly highlight the merits of a seasonal resolved reconstructions (Fig. 8). Another example is the Southeastern Australia drought, which had persistent dry conditions during the warm season for up to 10 years.



Natural variability and, in particular, the low-frequency contributions from decadal modes such as the IPO could contribute to those conditions in various ways. Additional realisations of multi-year droughts can help to understand physical mechanisms that lead to dry conditions, untangle dynamical interactions of various climatic modes, and lead to a better understanding of future drought risks. Inverse approaches and additional palaeoclimate reconstructions could help to deduce the climatic drivers causing these diverse drought characteristics and could provide important insights into the climate modes, their interactions, and influences on Australian droughts and pluvials. Our multi-century, seasonally and spatially resolved reconstructions provide new opportunities to study the dynamics of droughts across the Australian continent. For example, do similar settings such as a positive IOD, a positive ENSO, and a positive IPO lead to similar impacts on Australian droughts? It is imperative that we answer questions like this due to Australia's significant vulnerability to prolonged drought episodes.

**Acknowledgments**. M.B.F. and D.J.K. are supported by the Australian Research Council Centre of Excellence for Climate System Science (CE110001028). B.J.H. is supported through an Australian Research Council Linkage Project (LP150100062) and is an Associate Investigator of the Australian Research Council Centre of Excellence for Climate System Science. K.J.A. and P.J.B. are supported through an Australian Research Council Linkage Project (LP12020811).We thank the Bureau of Meteorology, the Bureau of Rural Sciences and CSIRO for providing the Australian Water Availability Project data. This is a contribution to the PAGES 2k Network. Past Global Changes (PAGES) is supported by the US and Swiss National Science Foundations. The authors thank Joelle Gergis, Jonathan Palmer, Ed Cook, Josephine Brown and Ailie Gallant for their generous advice on this project.



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





**Table 1. Summary of climate drivers, regions and droughts used in this study.** (a) Climate indices, (b) Natural Resource

Management (NRM) regions of Australia and (c) Instrumental and historical droughts

| (a) Climate Indices | | | (b) NRM Regions | | (c) Droughts | |
|---|---|---|---|---|---|---|
| Climate Index | Name | Ref | Region Abbrev. | Region name | Drought name | Period |
| SOI | Southern Oscillation Index | BOM | MN | Monsoonal North | Millennium Drought | 1997-2009 |
| NCT | Niño Cold Tongue Index | (Ren and Jin, 2011) | WT | Wet Tropics | World War II Drought | 1935-1945 |
| NWP | Niño Warm Pool Index | (Ren and Jin, 2011) | EC | East Coast | Federation Drought | 1895-1903 |
| EMI | El Niño Modoki Index | (Ashok et al., 2007) | CS | Central Slopes | SE Drought | |
| BLK | Blocking Index | (Pook and Gibson, 1999) | MB | Murray Basin | Goyder Line Drought | 1861-1866 |
| STRI | Subtropical Ridge Intensity | (Drosdowsky, 1993) | SSWF | Southern and Southwestern Flatlands | MD Basin Drought | 1797-1805 |
| STRL | Subtropical ridge Location | (Drosdowsky, 1993) | SS | Southern Slopes | Great Drought | 1809-1814 |
| DMI | Indian Ocean Dipole | (Saji et al., 1999) | R | Rangelands | Sturt Drought | 1809-1830 |
| | | | | | Black Thursday | 1849-1866 |
| | | | | | SE Drought | 1836-1845 |
| | | | | | Settlement Drought | 1790-1793 |



**Table 2. Extreme years**. Summary of the ten driest and wettest seasons for each NRM region for different baselines. Different baselines refer to the instrumental period (Instru: 1900-2014) and the extended reconstruction period (Pre-Instru: 1200-2014). Years highlighted in bold are among the ten-highest/lowest values for the entire reconstruction and instrumental period and therefore referred as extreme. Note the reconstruction period starts for verified periods only and differs for regions and seasons.

| Region | Extreme | Period | Warm Season | Cool Season |
|---|---|---|---|---|
| CS | Driest | Instru | **1902**, **1919**, **1930**, **1951**, **1941**, 1901, 1905, 1900, 1918, 1942 | **1994**, 1982, 2002, 1941, 1959, 1932, 1972, 1940, 1929, 1902 |
| | | Pre-Instru | **1433**, **1868**, **1791**, **1391**, **1431**, 1833, 1542, 1695, 1386, 1692 | **1896**, **1305**, **1607**, **1535**, **1623**, **1521**, **1569**, **1530**, **1380**, 1502 |
| | Wettest | Instru | **2011**, **1970**, **1962**, **1971**, **1950**, **1983**, **1910**, **2010**, **1974**, 1973 | **1998**, **1983**, **1920**, **1988**, **1990**, 1950, 1921, 1915, 1938, 1952 |
| | | Pre-Instru | **1644**, 1618, 1662, 1748, 1716, 1370, 1613, 1739, 1628, 1733 | **1557**, **1878**, **1796**, **1513**, **1745**, 1212, 1206, 1764, 1405, 1432 |
| EC | Driest | Instru | **1919**, **1942**, **1905**, **1945**, **1936**, 1901, 1937, 1902, 2006, 1992 | **1946**, **1918**, **2004**, **1994**, 1951, 1960, 1968, 1965, 1982, 1991 |
| | | Pre-Instru | **1386**, **1383**, **1413**, **1807**, **1428**, **1542**, **1391**, **1499**, **1474**, 1385 | **1800**, **1716**, **1681**, **1679**, **1871**, **1860**, 1760, 1714, 1680, 1758 |
| | Wettest | Instru | **2010**, **1970**, **1974**, **1971**, **1975**, **1973**, 1960, 1962, 1910, 1955 | **1983**, **1988**, **1989**, **1998**, **1931**, 1912, 1913, 1924, 1920, 1949 |
| | | Pre-Instru | **1752**, **1740**, **1732**, **1875**, 1627, 1731, 1602, 1753, 1743, 1742 | **1728**, **1820**, **1726**, **1769**, **1879**, 1770, 1786, 1742, 1787, 1833 |
| MB | Driest | Instru | **1902**, **1900**, **1905**, **1901**, **1931**, 1918, 1925, 1932, 1963, 1951 | **1982**, **1976**, **1994**, **1966**, **2006**, **1980**, 2002, 1925, 1936, 1914 |
| | | Pre-Instru | **1540**, **1691**, **1251**, **1695**, **1542**, **1485**, **1543**, **1394**, 1900, 1899 | **1778**, **1779**, **1811**, **1838**, 1480, 1817, 1885, 1481, 1607, 1780 |
| | Wettest | Instru | **2010**, **1992**, **2011**, **1950**, 1973, 1971, 1955, 1983, 1970, 1956 | **1915**, **1916**, 1955, 1973, 1956, 1970, 1974, 1968, 1975, 1917 |
| | | Pre-Instru | **1694**, **1668**, **1588**, **1751**, **1340**, **1750**, 1298, 1795, 1693, 1732 | **1572**, **1516**, **1706**, **1495**, **1649**, **1532**, **1523**, **1575**, 1537, 1721 |
| MN | Driest | Instru | **1951**, **1902**, **1905**, **1991**, **1935**, 1919, 1989, 1965, 1918, 1953 | **1926**, **1931**, **1930**, **1935**, **1932**, **1933**, 1994, 2002, 1964, 1934 |
| | | Pre-Instru | **1896**, **1761**, **1837**, **1838**, **1899**, **1814**, 1746, 1760, 1762, 1758 | **1745**, **1818**, **1684**, **1783**, 1878, 1667, 1658, 1760, 1808, 1900 |
| | Wettest | Instru | **2010**, **2000**, **1973**, **1999**, **2008**, **1950**, **1976**, **1998**, **1975**, **2003** | **2010**, **2006**, **1955**, **1910**, **1959**, **2000**, **1950**, **1956**, **1983**, 1974 |
| | | Pre-Instru | 1893, 1720, 1887, 1886, 1731, 1879, 1870, 1802, 1722, 1805 | **1694**, 1882, 1881, 1887, 1826, 1879, 1669, 1739, 1801, 1690 |
| R | Driest | Instru | **1965**, **1964**, **1905**, **2004**, **1951**, 1963, 1912, 1925, 1902, 1953 | **1925**, **1940**, **1976**, **1902**, **1994**, **2002**, 1946, 1926, 1941, 1944 |
| | | Pre-Instru | **1745**, **1746**, **1607**, **1872**, **1611**, **1696**, **1683**, **1698**, **1760**, 1868 | **1891**, **1832**, **1855**, **1812**, 1838, 1817, 1849, 1888, 1868, 1862 |
| | Wettest | Instru | **2010**, **1999**, **2000**, **2011**, **1979**, **1980**, **1973**, **1978**, **1976**, **1975** | **1998**, **2010**, **1974**, **1970**, **1978**, **1968**, **1973**, 1992, 1933, 1904 |
| | | Pre-Instru | **1664**, **1673**, **1886**, **1855**, 1720, 1690, 1735, 1672, 1663, 1633 | **1879**, **1825**, **1826**, 1819, 1829, 1870, 1881, 1880, 1861, 1894 |
| SS | Driest | Instru | **1919**, **1963**, **2006**, 1997, 1913, 2002, 2012, 1982, 1977, 1967 | **1940**, **1902**, **1982**, **1999**, **1966**, 2008, 1937, 1977, 1967, 1987 |
| | | Pre-Instru | **1439**, **1381**, **1437**, **1799**, **1438**, **1744**, **1868**, 1818, 1413, 1433 | **1855**, **1865**, **1888**, **1887**, **1817**, 1840, 1799, 1869, 1833, 1784 |
| | Wettest | Instru | **2010**, **1910**, **1955**, **1974**, **1950**, 1930, 1992, 1948, 1956, 1988 | **1961**, **1960**, **1974**, **1958**, **1975**, **1962**, **1973**, 1942, 1956, 1953 |
| | | Pre-Instru | **1890**, **1731**, **1398**, **1372**, **1894**, 1649, 1892, 1740, 1730, 1887 | **1789**, **1872**, **1848**, 1774, 1871, 1759, 1810, 1750, 1859, 1843 |
| SSWF | Driest | Instru | 1951, 1975, 1905, 1965, 1930, 1990, 1963, 2004, 1949, 1900 | **1957**, **2006**, **1914**, **1976**, **1940**, 1994, 1982, 2010, 2002, 1911 |
| | | Pre-Instru | **1814**, **1445**, **1610**, **1559**, **1536**, **1449**, **1223**, **1255**, **1247**, **1297** | **1514**, **1376**, **1481**, **1492**, **1488**, 1550, 1494, 1459, 1555, 1509 |
| | Wettest | Instru | **1999**, **1914**, **1916**, **1933**, **1938**, **2005**, **2011**, 1959, 1992, 1942 | 1931, 1915, 1917, 1909, 1907, 1927, 1908, 1910, 1942, 1920 |
| | | Pre-Instru | **1417**, **1565**, **1588**, 1340, 1427, 1586, 1245, 1649, 1218, 1239 | **1871**, **1603**, **1370**, **1470**, **1367**, **1605**, **1612**, **1759**, 1368, **1808** |
| WT | Driest | Instru | **1982**, **1905**, **1941**, **1925**, **1965**, 1902, 1904, 1968, 1991, 1946 | 1967, 1953, 1965, 1966, 1968, 1915, 2008, 1923, 1991, 1997 |
| | | Pre-Instru | **1314**, **1671**, **1251**, **1504**, **1313**, **1317**, **1761**, **1335**, 1305, 1433 | **1607**, **1251**, **1495**, **1391**, **1540**, **1584**, **1608**, **1536**, **1284**, **1223** |
| | Wettest | Instru | **2010**, **1975**, **1973**, **1974**, **1998**, 2000, 1976, 1971, 1910, 1917 | **2006**, **1989**, **1990**, 1976, 1983, 1981, 1956, 1945, 1971, 1972 |
| | | Pre-Instru | **1469**, **1752**, **1886**, **1467**, **1678**, 1242, 1425, 1241, 1874, 1471 | **1245**, **1231**, **1212**, **1210**, **1881**, **1661**, **1340**, 1413, 1228, 1209 |





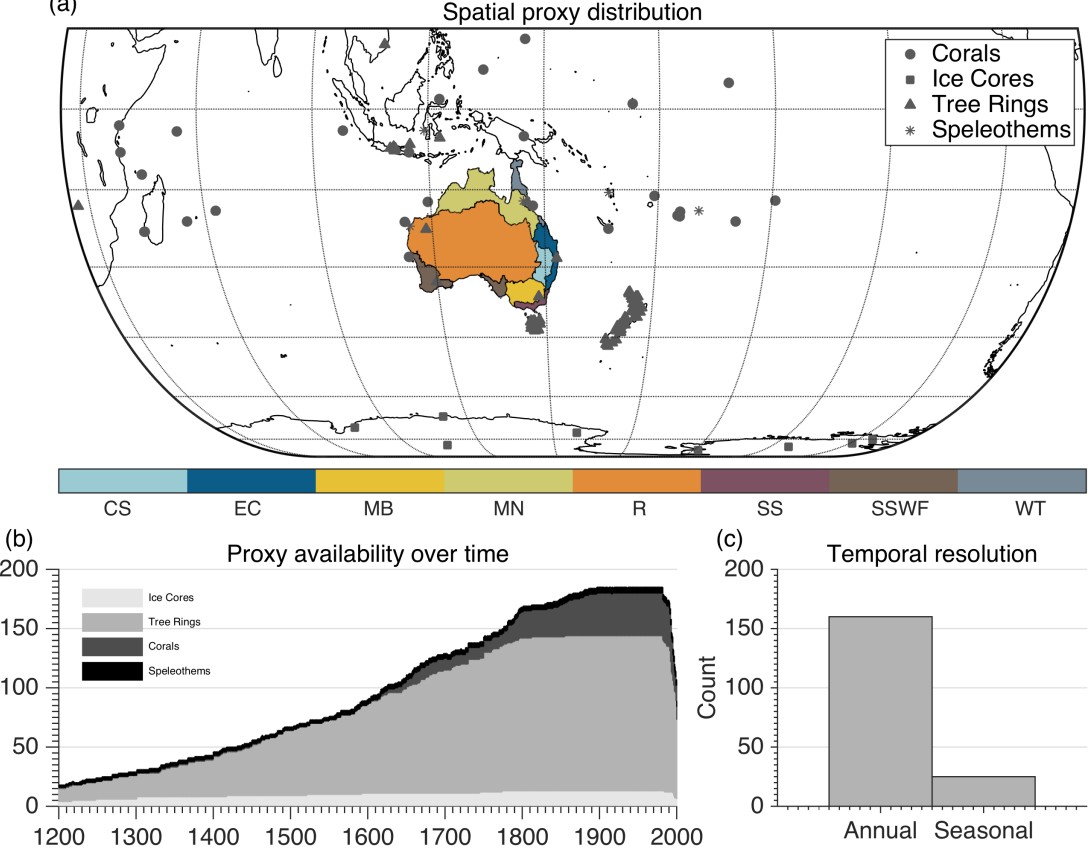

**Figure 1. Overview of Southern Hemisphere multi-proxy network.** a) Spatial distribution of 202 individual proxy records by archive type; National Resource Management (NRM) clusters shown on Australian continent. b) Record availability as a function of archive and time period covered, c) Temporal resolution of the palaeoclimate records



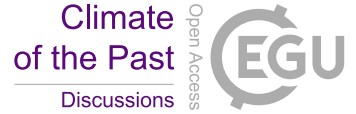

**Figure 2. NRM regions and their dominant climate influences on cool and warm season rainfall.**
Centre maps show the climate driver with the highest correlation to seasonal precipitation according to
the NRM regions. The drivers are summarised into four major categories: ENSO, IOD, SAM/BLK and
STR (See Table 1a)). Individual correlations between regional rainfall and each climate driver index are
5   given in surrounding bar plots. Only significant correlations exceeding the 10% significance level are
shown.



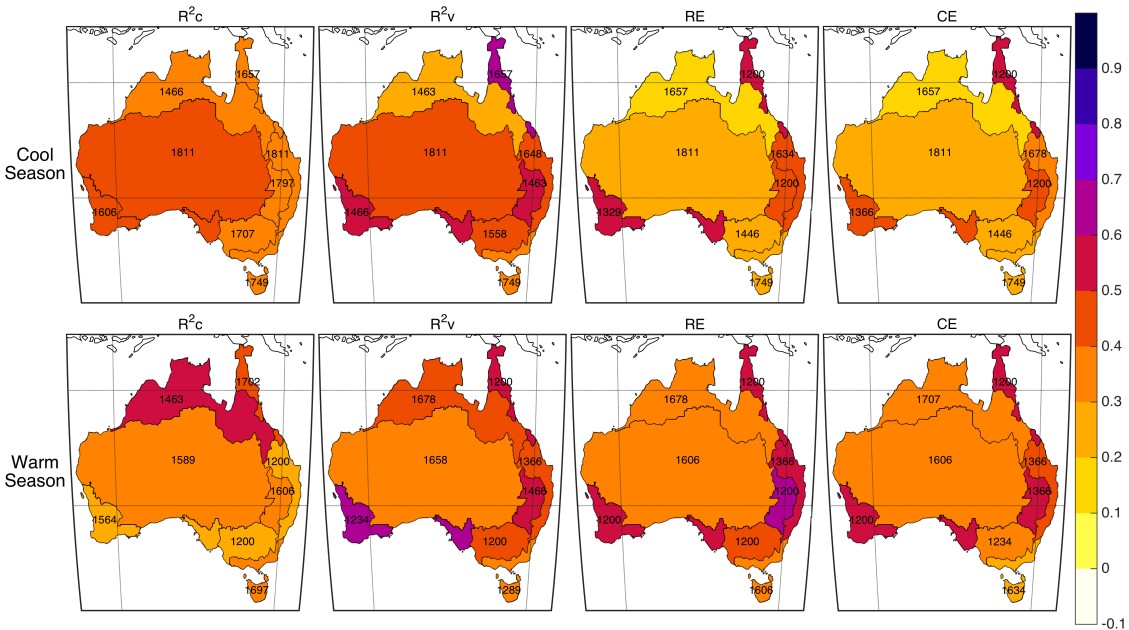

**Figure 3. Maps of calibration and verification statistics for the NRM regions.** Columns from left to

5  right are: Variance explained in the calibration period (R$^2$c), Variance explained in the verification

period (R$^2$v), Reduction of Error (RE) and Coefficient of Efficiency (CE) statistics for both the cool

season (top row) and the warm season (bottom row). Statistics shown apply to the most replicated nest

of the reconstruction. Numbers shown in each region indicate, for explained variance, the year in which

R$^2$c and R$^2$v reduce to half of their maximum value; and for RE and CE, the year in which RE and CE

10  remain positive, allowing for brief periods of negative skill of duration less than 5 years.

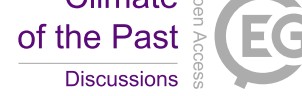

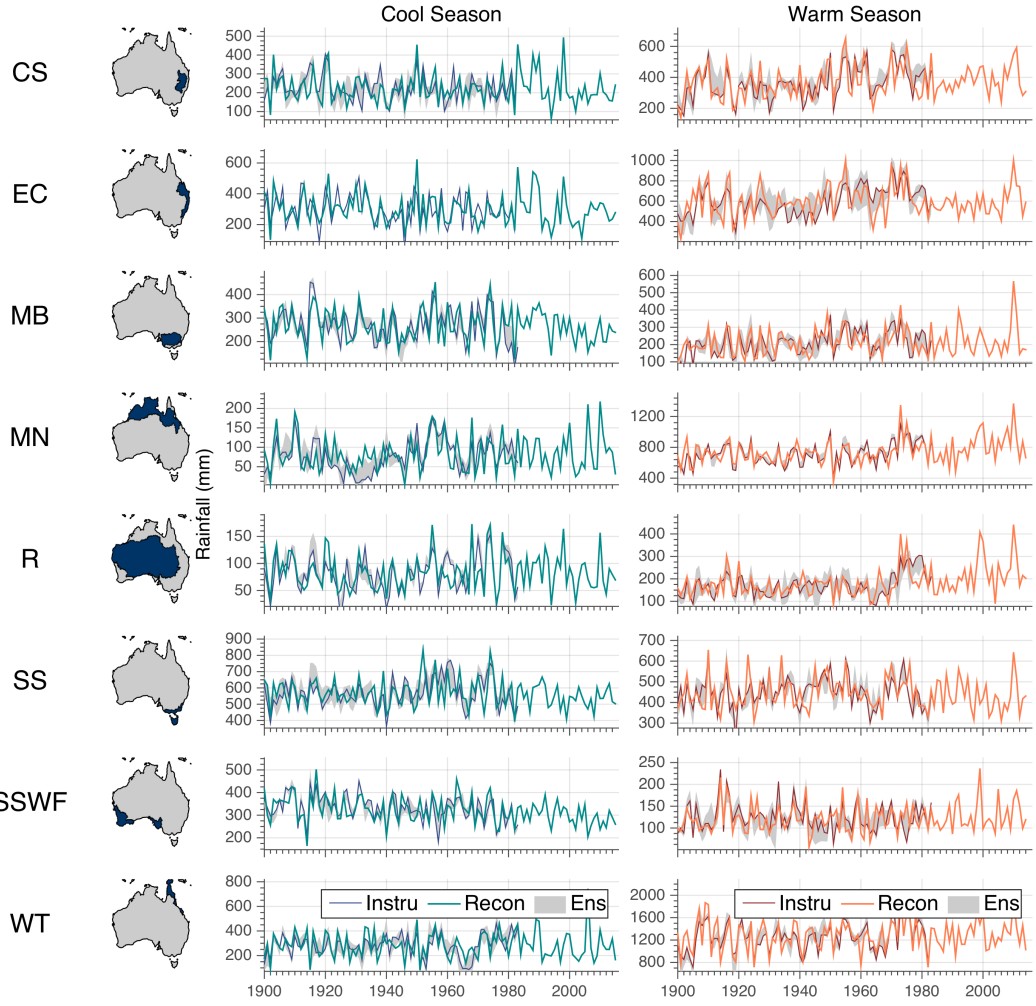

**Figure 4. Australian regional rainfall reconstructions during the instrumental period (1900-2015).**

Reconstructed cool (left) and warm (right) season rainfall is compared with the instrumental. Shaded in grey are uncertainty estimates based on the ensemble spread.







**Figure 5. Australian regional rainfall reconstructions in cool and warm seasons.** Regional

reconstructions for warm (red line) and cool season rainfall (blue line) from 1600- 1984, extended with

instrumental data from 1985-2015. Note, multiple axes (warm season rainfall according to left axis, cool

season refers to right axis). Shaded in grey are uncertainty estimates based on the ensemble spread. Red

5   and blue shaded periods indicate in phase and out-phase relationships between the seasons based on

windowed correlations (30-yrs)> 0.5.





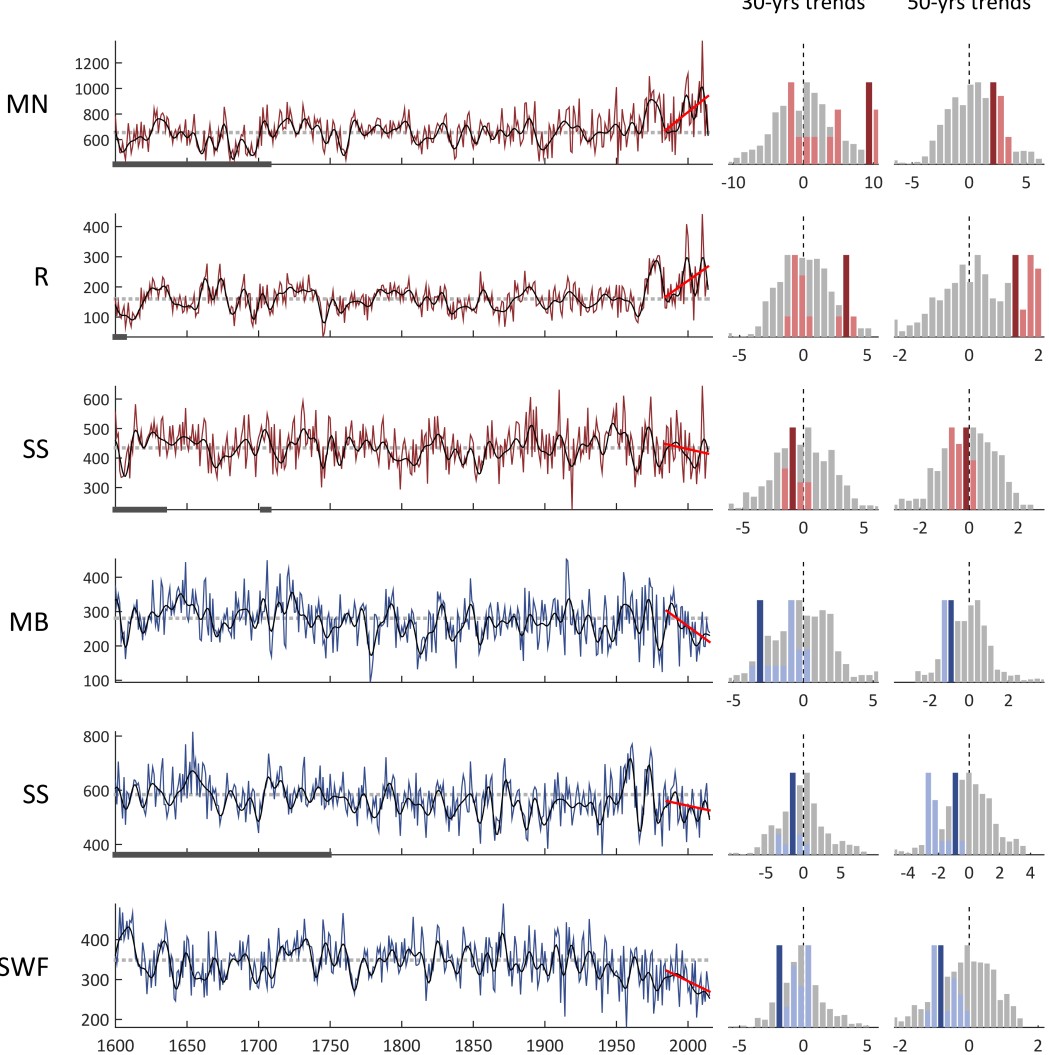

**Figure 6. Contextualising recent observed trends in regional Australian rainfall.** Left panels show

regional rainfall reconstructions since 1600 for the warm (red) and cool season (blue) with the 10-yr

low pass Chebyshev filtered series shown as a black line. Grey bars along the x-axis denote non-verified



periods for each reconstruction. Right panels show histograms of 30-yr and 50-yr regional rainfall

trends (mm/yr). Grey shaded bars indicate the full range of the trends prior to 1970 (for 30-year periods)

and 1950 for 50-year periods. Light red / blue colouring highlights the trends since 1970 (for 30-year

periods) or 1950 for 50-year periods. The dark coloured bars indicate the trend in the most recent

5    period. Bar heights are normalised by the maximum occurrence for each region.





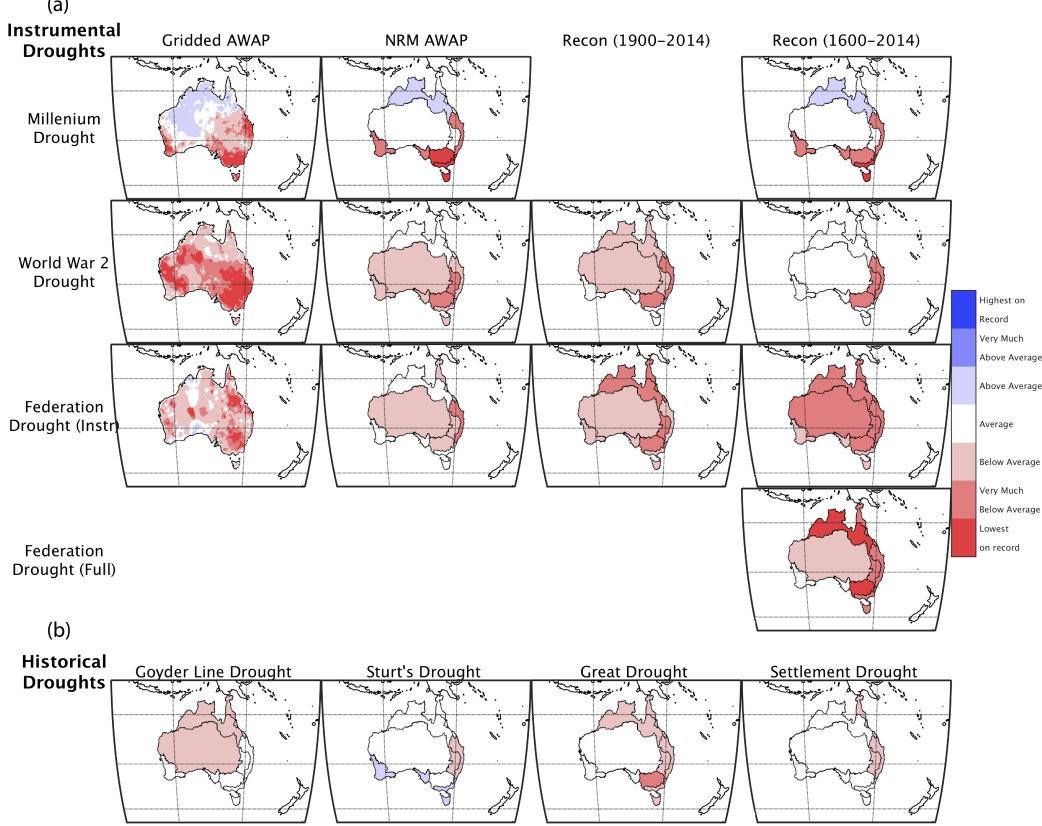

**Figure 7. Annual deciles for major droughts.** Plots for a) Significant drought periods during the instrumental period. Rankings of drought intensity are shown for three major instrumental period droughts, Column 1: AWAP gridded rainfall (1900-2014), column 2: NRM clusters (1900-2014), column 3: Regional reconstructions during instrumental period (1900-1990), column 4: Regional reconstructions during a four-century period (1600-2014). b) Rankings of major drought periods during the reconstruction period (1600-2014).





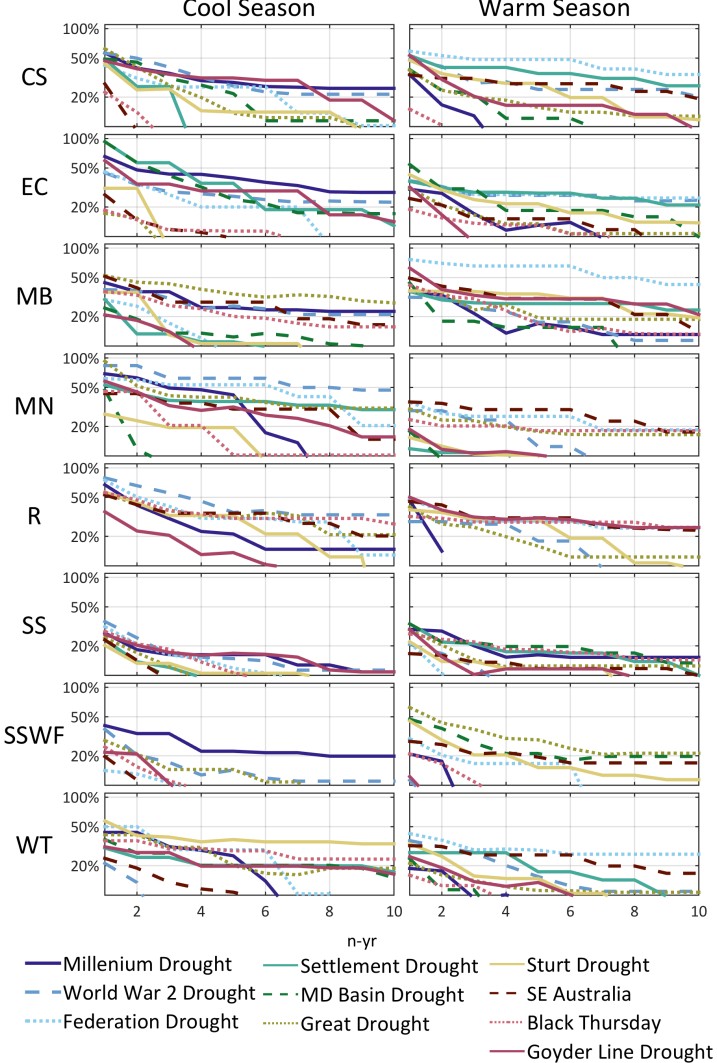

**Figure 8. Rainfall drought depth duration percentages across the regions**. The percentage reduction

below the long-term average of the driest years of variable duration (1-10 years) within a the selected

drought periods, for the cool season (left) and warm season (right).


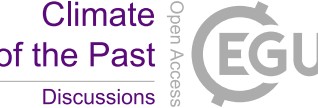


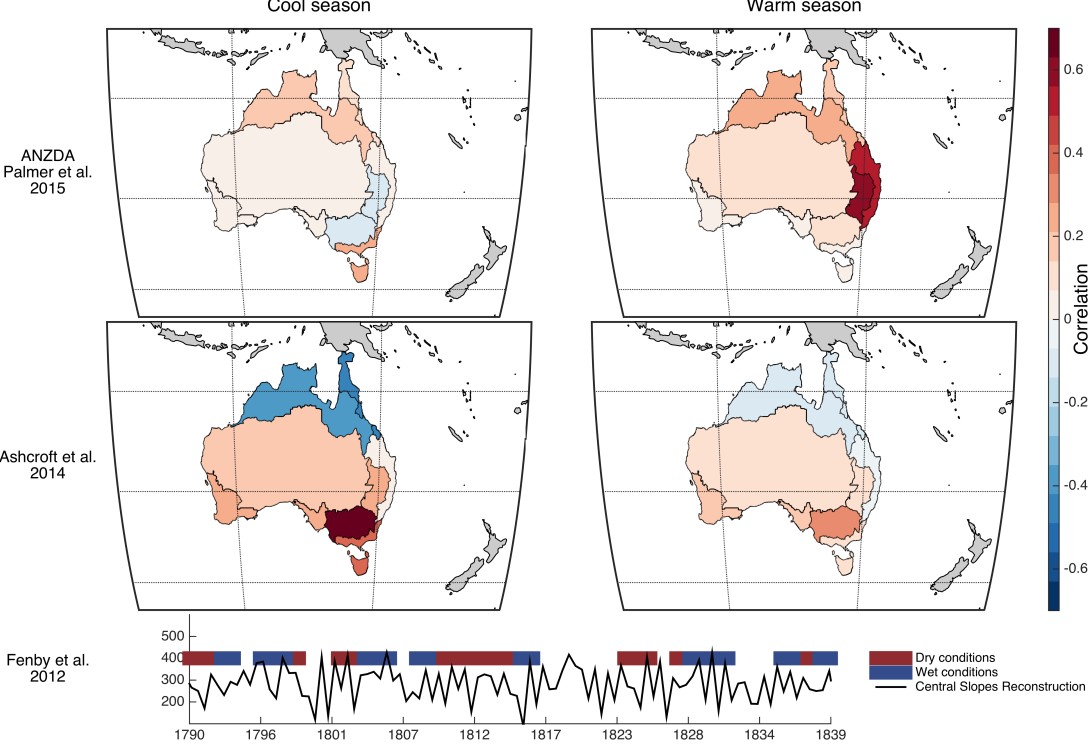

**Figure 9. Comparison with published studies.** Spatial correlation plot of regional NRM rainfall reconstruction and published studies on drought and rainfall. Spatial maps show region-wise correlation with the summer Australia New Zealand drought atlas (ANZDA) prior to instrumental period (1600-1899) (Palmer et al., 2015), season-wise correlation with high-quality observational data from southeast Australia (Ashcroft et al., 2014) from 1832-1859 and a quantitative/visual comparison with pre-instrumental documentary sources compiled by (Fenby and Gergis, 2012) for southeast Australia (Central Slopes reconstruction from this study).