# Peer review of "Multi-century cool and warm season rainfall reconstructions for Australia's major climatic regions"

_Climate of the Past, 2017_

## Referee Comment (RC1) · Anonymous Referee #1 · 20 Apr 2017

General Comments

This paper is generally good and well written. I have a few comments, most of which are relatively minor.

Major Comments

1. The abstract refers to cool and warm season rainfall reconstructions as being sub-annual. Elsewhere in the abstract they are referred to as seasonal and on p11 as bi-seasonal. Sub-annual is confusing. What they really are is half-year reconstructions, with the two parts of the year based on the cold and warm part of the year. Stick with one definition for them. Bi-seasonal seems best, sub-annual doesn't mean anything.

no2. When using CPS and rescaling using the mean and SD from the calibration period, you are assuming a normal distribution. How good is this for some of the smaller NRM regions? Particularly for the warm season, some regions show large positive spikes indicating that the distributions for some regions will be positively skewed. This makes using CPS more difficult, and it might be that proxies are less good at differentiating wet seasons from one another, but it could be worth a comment.

3. You've used deciles based on the period 1900-2014, so for the proxy series the last 30 years up to 2014 is based on instrumental data. Do your results get altered by basing both on the period from 1900-1984 and not padding with instrumental data? Also you do pad with 30 years, not the 20 you say, as 2014 is 30 years after 1984.

4. What would be useful in Table 1, when text on p9 discusses the cool and warm season rainfall series, is to add a % value for how much the cool or the wet season contributes to the overall 'annual' total. You could base this on a April to March year. This could help in the importance of some of the rainfall declines. Some regions get much more rain in one season compared to the other.

5. Is it worth also concluding that the 400-year reconstruction didn't produce a drought as extreme as the 3 in the instrumental period? As these are different lengths, did you go back and look at droughts of different lengths of numbers of years. The Millennium drought was 13, the WW2 drought 11 and the Federation drought 9.

Minor Comments

1. In the abstract or in the Introduction you mention the high-variability of Australian rainfall. This could be emphasized a bit more, as Australian averages (when expressed in percentage terms) are highly variable compared to other parts of the world. I recall seeing a plot of N and separately S Australian averages (Giorgi regions) compared to other similar sized regions of the world and Australia needed a different scale from all other regions.

2. On p2 lines 19-26 you talk about decrease in rainfall. Might be worth mentioning in impact terms that the costs of droughts are much more than the costs of floods. I'm assuming this is the case?

3. Add in on p3 that Cook has also produced the OWDA (Old World Drought Atlas) with a paper in 2016 in Science Advances.

4. Line 12, change Europe to Eurasia as there are lots of proxies across the whole boreal forest zone and from eastern Asia.

5. Useful if Figure 1 and Table 1 could be linked and the map named the 8 regions. It took me a while to realize that the big bit in the middle was called 'Rangelands'. I t also seems as though this region is just what's left from naming the other regions.

6. P4, line 16, these two references are missing (BJ93 and T et al.2015).

7. Another ref missing on p6 line 4. A better ref here would be Cook et al (1994, IJC, 14, 379-402).

8. On p6, the dates of the various droughts do overlap – maybe they are close regions and overall only affect part of Australia? Worth mentioning this though.

9. P7 introduces STRP and STRI, but Table 1 just refers to STRI and STRL. How do these two relate to STRP?

10. On p8 the cool season paragraph refers to 3 regions which extend back to 1200, 1260 and 1366. This is OK, but in the next paragraph the wet season 3 regions extend back to just 2 years?

11. Remove the 'as' before ENSO's on line 14 of p12.

12. Move the left bracket on line 7 of p13, so begin with Cook et al. (2016) found. . ..

13. In Table 1, SE Drought is in twice in the third column. This is why it doesn't get a date the first time? I presume all these dates are the accepted dates?

---

## Referee Comment (RC2) · Anonymous Referee #2 · 30 May 2017

With appropriate corrections, this paper will be a useful contribution to the literature related to the character and causes of variation in Australian hydroclimatology. Much of what is done is interesting, and the sub annual approach is great to see, but there needs to be some additional attention to detail, particularly related to the rationale and specifics of the research methodology, and a more critical approach to the results presented would be ideal. The paper has the potential to be very good, but I think it has some way to go to get there.

Below, I discuss aspects of the paper sequentially, explicitly highlighting what I consider critical points that I think must be addressed and major points that should be. Trivial points are collected at the end.

Abstract The abstract reads well, but minor changes will be required if the authors accept some of the criticisms that follow (e.g. the statement [p1, 15] that the rainfall reconstruction aligns well).

1 Introduction

[p2, 15]. State when instrumental data collection started. More generally, make sure that you are not assuming your readers are Australians when it comes to what may seem to be common knowledge.

[p2, 28–33]. This is a useful paragraph, but it would it be useful to extend it slightly with a comment on the relevance of palaeoclimate reconstructions under future conditions of changed boundary conditions.

Major. [p3, 11–17]. It might be useful to rephrase "process-based methodology" to more clearly capture the atmospheric dynamical aspect of what you are doing. Also, you need to explain why this approach will maximise skill and utility. If I recollect correctly, the advocates of the Cook approach of point-based regression would argue that this achieves the same. You need to justify your claim here.

Major. [p3, 11–17]. I was surprised to see the analysis based on NRM regions. The approach is contrary to what seems the more common and sophisticated approach of examining relationships at finer spatial resolution, so I would like to see some rationalisation for the choice here. A key criticism is that the spatial scale is too coarse for some regions to adequately capture the character of spatial hydroclimatological extremes and risks conflating contrasting regions into an unhelpful whole. My concerns here returned when I encountered Figure 4, where it is clear from the instrumental data that the regionalisation approach has some undesirable consequences. For the millennium drought, the bipolar R region pattern cancels out; for the WW2 drought, widespread drought in the west is lost; for the Federation drought, the centres of drought are displaced east. I do appreciate that you are not in a position to revise the analysis, but think you should give a more convincing rationale for the approach taken, and follow up

with a paragraph in the discussion to discuss the implications and outline if you think an alternative approach would be useful (or not).

[p3, 23]. The NRM regions cannot be clearly distinguished on Figure 1. Figure 4 is much better.

2.1 Instrumental data

[p4, 8–10]. Some expansion of the description of the AWAP data would be good. For example, it would be useful to state what homogeneity analysis has been undertaken (by BoM).

Major. [p4, 11–18; Table 1]. Insufficient information is provided on the climate drivers. For example, the metrics for the intensity and position of the subtropical ridge [over Australia] are not common knowledge, ditto blocking, and there are multiple indices for the SOI. All of this can be simply solved by adding an appropriate descriptor to Table 1. SAM appears to be missing from the table. The IPO is mentioned later but not used in the analysis and there is no equivalent west-pole Southern Oscillation index (you have one for SST, I presume that is what NWP is). Perhaps a little more rationalisation would be appropriate.

2.2 Palaeoclimate data

Critical. [Section 2.2]. Overall, Section 2 seems too superficial. The reader needs a better understanding of this fundamental data in order to interpret the subsequent results. See following for specific details.

Major. [p4, 20; Figure 1]. A cross-reference to details in the supplement is needed here. Also, the mapping is not up to the task of showing the spatial distribution (need zoomed in insets for high density areas) – e.g. I can only see one of the five speleothem proxy locations. It would also be useful to colour-code the symbols to show the spatial degradation back in time. Also, is it possible to distinguish those proxies actually used? Table S1 indicates numerous proxies that were not used for any region (all zeros).

Critical. [Missing details – proxy data pre-processing]. It is common practice to pre-process proxy data in ways that unavoidably affect the frequency response of any climate reconstruction. It appears (and you should state) that you do not re-process the data to ensure consistency, but it is essential that you comment on what has been done by the original workers (or subsequently). Without this information, your readers may incorrectly assume that Australian hydroclimatology is characterised by essentially no centennial-scale variability, when in fact the case is that it has been removed. Although a critical omission, the solution is very simple – you just need to state what frequency information is credible in the reconstruction. A related paragraph in the discussion would also be appropriate.

Critical. [Missing details – proxy dating fidelity]. Similar to the above, you are assuming that the dating of the proxies is accurate. That is fine, but a comment to the effect that dating is not revisited here may be appropriate. However, Table S1 indicates that you have used a number of non-annual proxies, yet I see no comment on how these are meaningfully included in an annual-resolution reconstruction. The rationale, the explicit methodology (interpolation?), and the implications should be mentioned.

3.1 Reconstruction

[p5, 3–15]. Good to see this focus on stationarity. Looking at only linear relationships and ignoring lag relationships is simplified but acceptable. But can something more be said about the interquartile range approach? i.e. where exactly does this come from and has it been tested for this purpose? I presume this analysis relates to the binary scores in Table S1 (the table caption does not provide the relevant information).

[p5, 9, and relevant to multiple other places]. Statistical significance is mentioned here for the first time. Why 0.1 (seems a fairly weak test) and how are significance levels adjusted for autocorrelation?

Critical. [p5, 17–24]. This section describes the reconstruction methodology. The credibility of the work rests on this, so the reader needs to thoroughly understand the

details of what has been done. There is not sufficient detail for me to be sure I completely follow what has been done. While it is appropriate to lean on other references for comprehensive treatment (but relevant cited important references are missing from the references) the onus is on the authors to present sufficient details here. The Tierney et al level of detail is a useful model in this context.

[p5, 24]. How spliced?

[p5, 27–28]. 52, 33 years. At face value 1934–1984 & 1900–1933 gives 51, 34. Missing something?

[p5, 30]. "...not entirely independent..." could be interpreted as mostly independent, which is incorrect.

3.2 Analysis

[p6, 7–9]. Rationale for this analysis? I don't know what normalized trends means in this context.

[p6, 14–16]. Detail redundant here (provided in Table 1).

[p6, 18]. Deciles need a time interval (e.g. 36 months).

[p6, 19–21]. Can this be rephrased for clarity?

4.1 Regional climate driver influences

[p7, 13]. ENSO "stands out" only in the warm season. The cool season map is mostly red, but this is misleading when the more nuanced bar graph results are considered. See later comments on Figure 2.

[p7, 14]. 44% < "most", so presumably you mean something else.

[p7, 24]. SSWF has the only warm season yellow (IOD) bar.

4.2.1 Reconstruction skill

[p8, 3, 5]. Figure 3 panel labels are given in the text, but are not shown on the figure.

Major. [p8, 5–8]. Some clarification of this earliest year comment is required. First, although it doesn't say so in the text, the Figure 3 caption indicates that statistics relate to calibration rather than verification statistics. Wouldn't the latter be better? Second, why is half the maximum calibration variance explained an appropriate metric here, rather than a fixed R2 threshold? Third, given that you can only assess skill based on comparing with observations, I assume that the early dates relate to how reduced data sets (corresponding to nests) perform against the instrumental data. If this is incorrect then some additional explanation is required. Whether correct or not, have you taken into account degraded proxy performance with time outside of the calibration/verification period? Loss of sample depth, and thus signal, is characteristic of the tree ring data, so there is more to reduced performance than simply the number of proxies. Probably nothing much you can do about this, except to note that the early dates will be inflated (too early), but to an unknown degree.

4.2.2 Reconstruction time-series

[p8, 22]. Probably best to delete "...and past centuries (Fig. 5)", because all comments in this paragraph relate to Figure 4.

[p8, 29]. Define "low-frequency". I am struck by the lack of it.

[p8, 26–27]. Perhaps I am missing something here, but doesn't your rescaling methodology force this? If so, then this is not a relevant comment.

Major. [p9, 15–27]. This is an interesting approach, but I am unconvinced by the interpretation. Because 30 years is a fairly short window, I suspect that analysis of serially-correlated random numbers may give similar results to what you see here. If so, then the patterns identified cannot realistically be interpreted in the manner done, although the conclusion would be the same. I am not convinced that it amounts to "...an additional verification measure".

**4.3.1 Contextualising recent rainfall trends**

[p10, 7–22]. Apart from the apples vs. oranges caveat (see discussion of Figure 6), this seems OK, but it does beg the question why 30/50 year trends are a key metric, rather than, say, 30/50 year means and variance. See previous comment about providing the rationale for this aspect of the methodology.

**4.3.2 Contextualising the spatial extent and intensity of past droughts**

[p10, 31–32]. Surely two droughts are not enough to make such a relatively bold statement, especially since the reconstruction gets the significance of the two droughts around the wrong way (gridded AWAP shows WW2 drought is more significant, but reconstruction indicates the Federation drought).

Major. [p11, 11–19]. Figure 7 is nice, but here are confusing elements to the results that require explanation. Recon (1900-214) shows central region (R) below average for both the WW2 and Federation droughts. Recon (1600-2014) has WW2 average and Federation very much below average. While I appreciate that deciles are a moving target, drilling down into the results is needed to make sense of what is going on. At face value, Recon (1600-2014) lacks credibility, because it relegates arguably the most significant drought of the instrumental record, based on the instrumental data, to relative insignificance!

**4.3.4 Extreme years in a long-term context**

[p12, 27–28]. This is pushing the envelope, but I am not convinced that you have actually shown that the reconstruction is actually up to this rather demanding task. I would need to see that the instrumental extreme years are captured in roughly the appropriate order.

[p13, 2–16]. Some of this material may be better in the discussion.

[p13, 18–28]. Results in this section have to be taken at face value because the tabled presentation is not well suited to "seeing" the claimed patterns.
**CPD**

4.6 Comparing our reconstruction with previously published

[p13, 33–34]. Please be explicit about the degree of overlap (%).

[p14, 4]. Is linear correlation against the PDSI appropriate? I don't recall if the PDSI scales linearly and it is also a water balance approach, so has significant memory. My point here is that you might be short changing yourselves by an overly simplistic inter-comparison.

[p14, 8–10]. I don't understand where you are going with this the last sentence. It reads like a criticism of the PDSI, but I suspect that is not your intention. The temperature dependence targets evaporation, making PDSI arguably a superior drought index. And the spatially unresolved parts presumably relates to the point-based approach, which is also arguably superior (you certainly have not convinced me otherwise).

Major. [p14, 10]. The poor warm season agreement with the PDSI analysis, except for one region, is quite alarming, especially the near-zero relationships in regions containing the cities where most Australians live. Given that this affects the perceived credibility of Australian drought reconstruction, it might be appropriate to follow up on this here, or in the discussion.

[p14, 14–17]. The cool season SE results are encouraging (water resources implications), but not so the dry season. Coupled with the poor agreement with the drought atlas, and the unconvincing relationship with the coastal records, I'm left doubting the credibility of the reconstruction.

5 Discussion and conclusion

[p15, 6]. "Eastern Australian" is too broad a phrase – agreement is much more spatially restricted. Personally, I think "high-level" is overselling things. Given that you are reconstructing the same thing (drought) from significantly similar data sets, I was expecting to see most variance in common, and you are well shy of that.

[p15, 7–8]. I suggest you limit the "compared well" comment to the cool season.

[p15, 11]. Interesting comment about highlighting the quality, because to me they highlighted the limitations.

[p15, 14]. This is a reasonable statement. But not picked up is some notable evolution in patterns for some regions. For example, MN & R in Figure 6 appear to have increased variability in the late 20th c. Is this real, or a splicing artefact?

[p15, 14–]. I remain unconvinced by this regression slope analysis approach. It can tell you about the rate of change and its significance, but is that really the important metric in terms of the process explanations you then mention? It also misses important cumulative impacts. For example, the SS and SSWF results show a cumulative decline to a mean substantially lower and with the most extreme droughts all relatively recent. MN and R show the reverse. A different type of analysis would be required to assess the significance of these changes.

Major. [p16, 2–3]. Comparison of instrumental vs. reconstructed trends can only reasonably be made with relevant caveats associated with the pre-processing of the palaeo data. Pre-processing has likely reduced supressed multi-decadal trends, so your histograms in Figure 6 will be pulled in at the tails, which clearly will affect your assessment of how the instrumental data trends (which have not been similarly treated) compare. Note though that recognising this actually reinforces your conclusion about recent trends being within the range of natural variability.

Major. [p16, 9–10]. The discussion in this paragraph follows on and emphatically restates earlier comments about the quality of the reconstruction of historical droughts that I think can reasonably be challenged (see [p11, 11–19], above). First, surely you only have two droughts. The millennium drought is outside your proxy data period, so it is essentially spliced instrumental data, is it not? If so, then agreement of spatially-averaged instrumental data with the original gridded data is meaningless, although it does point to issues with spatial units that are too large (a paragraph discussing this spatial scaling issue would be appropriate). For the other two droughts, you can

only really claim good agreement for the Federation drought. As previously stated, I think the WW2 drought reconstruction is severely awry, and suggests to me that the methodology may only be suitable for capturing some types of drought (perhaps some additional forcings are not captured by the proxy network). The credibility of the reconstruction is challenged by the poor representation of what the instrumental data shows to be the most extreme drought in the instrumental period (Figure 7, left column).

[p16, 27–33]. This is an interesting point. Can you relate it back to the drivers?

Major. [p17, 3–4]. I don't disagree with this, but it does presuppose that teleconnection patterns will remain stable in a future warming world. The flip side of this is that the reconstructions extend back into a globally cooler period. If teleconnection were different then (and there is evidence to suggest they were for some of your proxies), what then are the implications for your reconstruction (because the transfer function will be wrong)? Moreover, drought is not just rainfall. Australian researchers have shown that droughts have intensified in response to increasing T (and thus evaporation), have they not? So a rainfall-only analysis is only part of the story. Surely worth some serious commentary.

Table 1. SAM is missing. Additional details of indices would be useful (e.g. I assume NCT and MWP are SST based). A sentence or two describing each index would be useful. Surprising you have not included a west pole pressure index.

Table 2. The caption could usefully be reworded for clarity. Is the information for "Instru" the reconstructed data for the instrumental period, the same but with instrumental data spliced on the end, or the instrumental data? This seems a rather ineffectual way of presenting the information – visualisation would highlight temporal patterns, temporal clustering, and inter-regional patterns in a way that tabled numbers do not.

Figure 1. See previous comment about inability to resolve the proxies and the regions on this map. Also, given that many proxies were eliminated, would it not be more useful

to limit the map to those proxies that actually end up being used. Moreover, it would be interesting to see this broken down by region in the supplement. Without a laborious process of extracting the relevant information from Table S1 and remapping, this useful information about contributing proxies is unavailable to the reader.

Figure 2. It appears that the SOI is generally superior or comparable to the other three ENSO indices. That point could be made in the text and this figure simplified. There is a wealth of information in the bar charts, but the maps are unsophisticated in the treatment of this, and I think counterproductive in oversimplifying matters. I don't recall comment in the text related to the logic of pooling SAM and BLK. It would be useful to include the region codes along with their long names on the bar charts (also on the maps). Consider adding a horizontal line separating the two parts of the figure. Because you don't have axis labels, you need to explicitly state in the caption that the bar graphs show correlations. See previous comment about uncertainty about whether autocorrelation has been allowed for in the significance levels cited (see [5, 9] above).

Figure 3. It would be useful to have the years corresponding to the plotted statistics shown.

Figure 4. There are several instances where the reconstruction is outside the ensemble range. Having gone to the trouble of calculating the ensembles (a good thing), why isn't a mean/median (or other measure of central tendency) used for the reconstruction? Doing so would "fix" some of the points of difference with the instrumental data (e.g. in MB, MN, WT). It would introduce other issues, but the net benefit may be positive, and a transfer function based on the full data rage may be more robust. Just a thought.

Figure 7. I presume that the millennium drought is "missing" for Recon (1900–2014), because you only have instrumental data. If that is true, I don't get why it appears in Recon (1600–2014) – its not reconstructed, its spliced instrumental data isn't it? Need to specify the time periods for decile calculation (12/24/36 months?).

Figure 8. Please expand the caption to better explain exactly what is being shown here.

How is the starting point for each drought determined?

Minor points

[p2, 8]. Delete "a" at the end of the line.

[p2, 10]. This style of referencing with a list of references at the end of a paragraph is unfortunate. They presumably don't all relate to the last point, and if they do then there are missing references in the body of the paragraph.

[p3, 27]. Do you mean "Compile"?

[p10, 32]. New paragraph (millennium drought)?

[p11, 11]. [somewhat] similar?

[p11, 33]. provide[s] insight.

[p12, 3]. Suggest you change "many" to "several"? 4–5/8 and only cool season.

[p12, 14]. Seems to [be] a result.

[p12, 24]. Breaking up paragraph into smaller ones would help readability.

[p12, 26]. Expand "Black Thursday" for benefit of non-Australian readers.

[p14, 5]. Reconstruction[s]?

[Table 2, 4]. referred [to] as.

[Figure 8, 3]. Delete "a" (3rd last word).

[Figure 9]. Fenby et al should be Fenby and Gergis.

---

## Short Comment (SC1) · 3 Jun 2017

D.s Kaufman

darrell.kaufman@nau.edu

The PAGES Data Stewardship Integrative Activity seeks to advance best practices for sharing data generated and assembled as part of all PAGES-related activities. As part of this activity, a team of reviewers has been constituted for the "Climate of the Past 2000 years" Special Issue. The data team is reviewing the data handling within each of the CP-Discussion papers in relation to the CP data policy and current best practices. The team has identified essential and recommended additions for each paper, with the goal of achieving a high and consistent level of data stewardship across the 2k Special Issue. We recognize that an additional effort will likely be required to

meet the high level of data stewardship envisaged, and we appreciate the dedication and contribution of the authors. This includes the use of Data Citations (see example in supplement). We ask authors to respond to our comments as part of the regular open interactive discussion. If you have any questions about PAGES Data Stewardship principles, please contact any of us directly.

Best wishes for the success of your paper,

2k Special Issue Data Review Team (Darrell Kaufman, Nerilie Abram, Belen Martrat, Raphael Neukom, Scott St. George) and ex-officio team members (Marie-France Loutre, Lucien von Gunten)

Essential additions for this paper:

(1) Add a "Data Availability" section to include a URL/Data Citation to a landing page that lists the datasets used in this paper (Table S1) and the URL/Data Citation for the primary output of this study.

(2) Add Data Citations or URLs (in addition to publication citations) for each of the 185 records used in for the rainfall reconstructions in this study. For those records not already in a persistent public repository, submit the essential metadata along with the proxy data and add the corresponding Data Citation (or URL) in Table S1.

(3) Submit the primary outcome of this study, the regional rainfall reconstructions for cool and warm seasons (Fig 5), to a public repository and include the Data Citation in "Data Availability".

(4) Archive the instrumental time-series targets for the reconstructions (Fig 4) along with the reconstructions.

Please also note the supplement to this comment:
http://www.clim-past-discuss.net/cp-2017-28/cp-2017-28-SC1-supplement.pdf

---

## Author Comment (AC1) · 7 Jul 2017

This paper is generally good and well written. I have a few comments, most of which are relatively minor.

Reply: We want to thank Anonymous Referee #1 for his/her careful and constructive review of our paper. We will take into account all his/her valuable advice for the revision of the manuscript. The reviewer's comments are in black. Below are our replies to his/her comments in blue. Comments in italic are suggested changes to the manuscript.

Major Comments

1. The abstract refers to cool and warm season rainfall reconstructions as being sub- annual. Elsewhere in the abstract they are referred to as seasonal and on p11 as bi- seasonal. Sub-annual is confusing. What they really are is half-year reconstructions, with the two parts of the year based on the cold and warm part of the year. Stick with one definition for them. Bi-seasonal seems best, sub-annual doesn't mean anything.

We agree with the reviewer and see that multiple definition could lead to confusion. We will change all references to: bi-seasonal.

2. When using CPS and rescaling using the mean and SD from the calibration period, you are assuming a normal distribution. How good is this for some of the smaller NRM regions? Particularly for the warm season, some regions show large positive spikes indicating that the distributions for some regions will be positively skewed. This makes using CPS more difficult, and it might be that proxies are less good at differentiating wet seasons from one another, but it could be worth a comment.

We agree with the reviewer and add a plot to the supplement showing the individual distributions. Most of the seasonal and regional rainfall distributions tend to follow a normal distribution. All instrumental cool season time series are approximate normally distributed (Chi-square test, 5% significance level). The distribution of instrumental rainfall during the warm season in the Central Slopes, Rangelands and Southern Slopes doesn't follow a normal distribution (Chi-square test, 5% significance level) and shows positive skewness (see Supporting Figure 1 for details).

We add a statement as followed p.5 l.24:

*"It should be noted that not all of the records strictly follow a normal distribution."*

3. You've used deciles based on the period 1900-2014, so for the proxy series the last 30 years up to 2014 is based on instrumental data. Do your results get altered by basing both on the period from 1900-1984 and not padding with instrumental data? Also you do pad with 30 years, not the 20 you say, as 2014 is 30 years after 1984.

We agree with the reviewer. Deciles strongly depend on the baseline. We added a plot to the supplement showing different baselines (S.Figure 2) and refer to it on page 6 line 22:

*"It is important to note that deciles deliver quantitative statements only in conjunction with a baseline period and duration (see details in supplementary figure 2)."*

4. What would be useful in Table 1, when text on p9 discusses the cool and warm season rainfall series, is to add a % value for how much the cool or the wet season contributes to the overall 'annual' total. You could base this on a April to March year. This could help in the importance of some of the rainfall declines. Some regions get much more rain in one season compared to the other.

We agree with the reviewer and add the contribution of seasonal rainfall in percentages to the annual average in table 1.

5. Is it worth also concluding that the 400-year reconstruction didn't produce a drought as extreme as the 3 in the instrumental period? As these are different lengths, did you go back and look at droughts of different lengths of numbers of years. The Millennium drought was 13, the WW2 drought 11 and the Federation drought 9.

Yes, this comment is true based on the 3 instrumental droughts we have been looked at. This is again true only in conjunction with a baseline period and duration of these droughts. Looking at other past drought events of different duration and intensity hasn't been done in this study and remains for future work.

Minor Comments

1. In the abstract or in the Introduction you mention the high-variability of Australian rainfall. This could be emphasized a bit more, as Australian averages (when expressed in percentage terms) are highly variable compared to other parts of the world. I recall seeing a plot of N and separately S Australian averages (Giorgi regions) compared to other similar sized regions of the world and Australia needed a different scale from all other regions.

Good point, we added the citation: {Nicholls:1997gh} to p.2 line 2.

2. On p2 lines 19-26 you talk about decrease in rainfall. Might be worth mentioning in impact terms that the costs of droughts are much more than the costs of floods. I'm assuming this is the case?

Not necessarily and depends on multiple factors. It is hard to quantify the costs of natural disasters. For example, the total economic cost of Queensland floods is estimated as $14bn (http://australianbusinessroundtable.com.au/our-papers/social-costs-report) compared to $7bn caused by Black Saturday bushfires. A report by the WMO estimated the cost of the 1981 drought with US$15.15bn (http://www.wmo.int/pages/prog/drr/ transfer/2014.06.12-WMO1123_Atlas_120614.pdf). It is hard to integrate and compare those costs and beyond the scope of this study, we therefore back away from any statement.

3. Add in on p3 that Cook has also produced the OWDA (Old World Drought Atlas) with a paper in 2016 in Science Advances.

Yes, we will add this reference!

4. Line 12, change Europe to Eurasia as there are lots of proxies across the whole boreal forest zone and from eastern Asia.

Yes, we have changed that.

5. Useful if Figure 1 and Table 1 could be linked and the map named the 8 regions. It took me a while to realize that the big bit in the middle was called 'Rangelands'. I t also seems as though this region is just what's left from naming the other regions.

Yes, we add this into the Figure caption 1:

*"Full names of the Natural Resource Management (NRM) clusters shown on Australian continent as abbreviations are given in table 1."*

6. P4, line 16, these two references are missing (BJ93 and T et al.2015).

We will add those.

7. Another ref missing on p6 line 4. A better ref here would be Cook et al (1994, IJC, 14, 379-402).

We will add this missing reference.

8. On p6, the dates of the various droughts do overlap – maybe they are close regions and overall only affect part of Australia? Worth mentioning this though.

Those droughts are historical reported droughts from different regions. Back in those days, they might have affected the entire continent, but were only reported in the settlement areas along the coast but those historical droughts seem to be have been indeed quite regionally constrained to a certain area. Overlapping periods will be definitely be future work but we will show individual years in an animation/video.

9. P7 introduces STRP and STRI, but Table 1 just refers to STRI and STRL. How do these two relate to STRP?

We will correct that in table 1.

10. On p8 the cool season paragraph refers to 3 regions which extend back to 1200, 1260 and 1366. This is OK, but in the next paragraph the wet season 3 regions extend back to just 2 years?

Good point, we changed in to p.8, line 15-16:

*"In the warm season, the Southern and Southwestern Flatlands, Wet Tropics, and Murray Basin warm season reconstructions are skillful (CE > 0) for the longest period, extending back to 1200,1200 and 1234, respectively "*

11. Remove the 'as' before ENSO's on line 14 of p12.  12. Move the left bracket on line 7 of p13, so begin with Cook et al. (2016) found. . ..

Will be changed.

13. In Table 1, SE Drought is in twice in the third column. This is why it doesn't get a date the first time? I presume all these dates are the accepted dates?

We will correct that in table 1.

[Figure]

*Supporting Figure 2: Normalised probability density distribution of the instrumental records (1900-1994), the reconstructed records during the instrumental period (1900-1994) and through the entire reconstructed period (1600-1900). Cool season records are shown in blue, warm season records are shown in red. Note all cool season records follow approximate a normal distribution according to a Chi-square test (5% significance level) while warm season records of Central Slopes, Rangelands and Southern Slopes are not normally distributed.*

---

## Author Comment (AC2) · 7 Jul 2017

RC2: With appropriate corrections, this paper will be a useful contribution to the literature related to the character and causes of variation in Australian hydroclimatology. Much of what is done is interesting, and the sub annual approach is great to see, but there needs to be some additional attention to detail, particularly related to the rationale and specifics of the research methodology, and a more critical approach to the results presented would be ideal. The paper has the potential to be very good, but I think it has some way to go to get there. Below, I discuss aspects of the paper sequentially, explicitly highlighting what I consider critical points that I think must be addressed and major points that should be. Trivial points are collected at the end. Abstract The abstract reads well, but minor changes will be required if the authors accept some of the criticisms that follow (e.g. the statement [p1, 15] that the rainfall reconstruction aligns well).

Reply: We want to thank Anonymous Referee #2 for his/her careful and constructive review of our paper. Given comments and suggestions are extremely helpful to clarify our manuscript. We will take into account all his/her advice for the revision of the manuscript. The reviewer's comments are in black. Below are our replies to his/her comments in blue. Comments in italic are suggested changes to the manuscript.

1 Introduction

[p2, 15]. State when instrumental data collection started. More generally, make sure that you are not assuming your readers are Australians when it comes to what may seem to be common knowledge.

Good point, we added this information in line 15.

*"Over the 20th century many regions in Australia have experienced prolonged pluvial and drought periods that are documented in the gridded, instrumental records starting in 1900."*

[p2, 28–33]. This is a useful paragraph, but it would it be useful to extend it slightly with a comment on the relevance of palaeoclimate reconstructions under future conditions of changed boundary conditions.

We agree and added the following statement to line 33.

*"Palaeoclimate data can provide a unique window into long-term rainfall variability and emerging spatial and temporal trends. Such knowledge has practical applications for water resources management, seasonal forecasting, future climate predictions and constraints on boundary conditions."*

Major. [p3, 11–17]. It might be useful to rephrase "process-based methodology" to more clearly capture the atmospheric dynamical aspect of what you are doing. Also, you need to explain why this approach will maximise skill and utility. If I recollect correctly, the advocates of the Cook approach of point-based regression would argue that this achieves the same. You need to justify your claim here.

We agree and clarified this aspect to a more dynamical-focused methodology. In contrast to the point-by-point regression approach by Cook, our reconstruction is constrained by dynamical processes rather than a certain distance to the target in order to allow for remote proxy to be possible predictors.

*"We utilize a more dynamically-focused methodology driven by dynamical relationships to include remote proxies and maximize the skill and widespread utility of our reconstructions of Australian rainfall"*

Major. [p3, 11–17]. I was surprised to see the analysis based on NRM regions. The approach is contrary to what seems the more common and sophisticated approach of examining relationships at finer spatial resolution, so I would like to see some rationalisation for the choice here. A key criticism is that the spatial scale is too coarse for some regions to adequately capture the character of spatial hydroclimatological extremes and risks conflating contrasting regions into an unhelpful whole. My concerns here returned when I encountered Figure 4, where it is clear from the instrumental data that the regionalisation approach has some undesirable consequences. For the millennium drought, the bipolar R region pattern cancels out; for the WW2 drought, widespread drought in the west is lost; for the Federation drought, the centres of drought are dis- placed east. I do appreciate that you are not in a position to revise the analysis, but think you should give a more convincing rationale for the approach taken, and follow up with a paragraph in the discussion to discuss the implications and outline if you think an alternative approach would be useful (or not).

We concentrate on larger regions that are developed by the Department of Environment as the eight natural resource management clusters and maximize its comparability and usability in connection with the reports by Climate Change in Australia to extend the records back in time. We compromise our analysis to a full spatial picture of hydroclimate variability over Australia (which isn't provided by the ANZDA) and concentrate on a smaller temporal scale rather than spatial scale. This coarse resolution compromises between available proxy data and the quality of instrumental data too.

We added a statement about discrepancies between the patterns reflected by the regions and gridded observations in the data section line 9.

*"The Climate Change in Australia report (CSIRO and Bureau of Meteorology, 2015) applied a regionalisation scheme to define eight Natural Resource Management (NRM) regions with similar climatic and biophysical features. These regions reflect broad pattern of large-scale rainfall variability but may not capture finer scale patterns".*

In the discussion (p.17 line 6) we added possible future work to emphasize the need of finer resolution to resolve important features as followed:

*"Our multi-century, seasonally and spatially resolved reconstructions provide new opportunities to study the dynamics of meteorological droughts across the Australian continent. Future work should consider further sub-division to resolve finer scale hydroclimate patterns important for regional assessments. This could include a sub-division of large regions such as the Rangelands and regions of complex rainfall regimes like Tasmania as suggested by the NRM sub-clusters (CSIRO and Bureau of Meteorology, 2015)."*

[p3, 23]. The NRM regions cannot be clearly distinguished on Figure 1. Figure 4 is much better.

We agree, will clarify Figure 1 and refer the reader to Figure 4.

2.1 Instrumental data

[p4, 8–10]. Some expansion of the description of the AWAP data would be good. For example, it would be useful to state what homogeneity analysis has been undertaken (by BoM).

We agree and added this statement to p.4 line 8:

*"The monthly AWAP dataset based on precipitation anomalies generated from a varying number of station observations using the Barnes successive-correction* (KOCH et al., 1983) *and a three-dimensional smoothing spline interpolation* (HUTCHINSON, 1995).*"*

Major. [p4, 11–18; Table 1]. Insufficient information is provided on the climate drivers. For example, the metrics for the intensity and position of the subtropical ridge [over Australia] are not common knowledge, ditto blocking, and there are multiple indices for the SOI. All of this can be simply solved by adding an appropriate descriptor to Table 1. SAM appears to be missing from the table. The IPO is mentioned later but not used in the analysis and there is no equivalent west-pole Southern Oscillation index (you have one for SST, I presume that is what NWP is). Perhaps a little more rationalisation would be appropriate.

We thank the reviewer and agree that important information is missing in Table 1. SAM was missing and has been added to Table 1. Details about the computation of the individual climate indices is added at two points. First, we added a reference within the instrumental data section in p. 4 line 11:

*"We also use several climate indices to link climate drivers with Australian rainfall (Table 1(a)) that have previously been used to characterize the relationship between rainfall and large-scale drivers (Risbey et al., 2009). Details on the computation of the individual climate indices follow strictly the metrics described in the references given in Table 1 and references within."*

Secondly, we corrected our table description to refer to the original publications within Table 1 and add the following descriptor to the table:

*"Summary of climate drivers, regions and droughts used in this study. (a) Climate indices and reference for computational information, (b) Natural Resource Management (NRM) regions of Australia and (c) Instrumental and historical droughts."*

Furthermore, the role of the position and strength of subtropical ridge has been highlighted in (Fiddes and Timbal, 2016; Timbal and Drosdowsky, 2012; Timbal et al., 2006). We cite those publications in line 31-32 and would refer the reader to those publications for further details. The role of blocking has been highlighted by (Pook and Gibson, 1999; Risbey et al., 2009)undertaking similar analysis during the instrumental period only. The SOI index partially incorporates information about west-pole conditions of the atmosphere by the accounting for pressure differences over Darwin. We didn't include the IPO in our analysis as we are focusing on year-to-year variability only. Given our approach of the moving correlation window of 30 years, decadal- variability such as expressed by the IPO has been not considered.

2.2 Palaeoclimate data

Critical. [Section 2.2]. Overall, Section 2 seems too superficial. The reader needs a better understanding of this fundamental data in order to interpret the subsequent results. See following for specific details.

Major. [p4, 20; Figure 1]. A cross-reference to details in the supplement is needed here. Also, the mapping is not up to the task of showing the spatial distribution (need zoomed in insets for high density areas) – e.g. I can only see one of the five speleothem proxy locations. It would also be useful to colour-code the symbols to show the spatial degradation back in time. Also, is it possible to distinguish those proxies actually used? Table S1 indicates numerous proxies that were not used for any region (all zeros).

We agree, and add an inset to Figure 1 showing the spatial distribution and degradation back in time following a color-code and add a cross-reference to the supplement.

Critical. [Missing details – proxy data pre-processing]. It is common practice to pre- process proxy data in ways that unavoidably affect the frequency response of any climate reconstruction. It appears (and you should state) that you do not re-process the data to ensure consistency, but it is essential that you comment on what has been done by the original workers (or subsequently). Without this

information, your readers may incorrectly assume that Australian hydroclimatology is characterised by essentially no centennial-scale variability, when in fact the case is that it has been removed. Although a critical omission, the solution is very simple – you just need to state what frequency information is credible in the reconstruction. A related paragraph in the discussion would also be appropriate.

*We agree with the reviewer and add this to the data section p.4 line 28:*

*"No further data treatment has been applied other than suggested by the original publication which involves the removal of non-climatic biological trends in tree-ring records using the signal-free method preserving much of the medium-frequency variability (timescales of decades to a century). (see S. Table for references and details)"*

Critical. [Missing details – proxy dating fidelity]. Similar to the above, you are assuming that the dating of the proxies is accurate. That is fine, but a comment to the effect that dating is not revisited here may be appropriate. However, Table S1 indicates that you have used a number of non-annual proxies, yet I see no comment on how these are meaningfully included in an annual-resolution reconstruction. The rationale, the explicit methodology (interpolation?), and the implications should be mentioned.

*Good point, sub-annually resolved records were binned into seasonal averages. We added to the data section p4. Line 31:*

*"Samples within the seasonal window (six consecutive months) are equality averaged onto a regular time grid of 2 samples a year, whereas the dating of annually resolved records follows the original author. "*

3.1 Reconstruction

[p5, 3–15]. Good to see this focus on stationarity. Looking at only linear relationships and ignoring lag relationships is simplified but acceptable. But can something more be said about the interquartile range approach? i.e. where exactly does this come from and has it been tested for this purpose? I presume this analysis relates to the binary scores in Table S1 (the table caption does not provide the relevant information).

*The interquartile range approach has been used as a straightforward indicator of stationarity. To the knowledge of the authors this approach hasn't been used anywhere else in this context. Several studies have shown that the relationships of Australian rainfall and climate drivers such as ENSO have periods of varying strength of the teleconnection (e.g. (King et al., 2014; Lewis and LeGrande, 2015)). The interquartile range ensures the teleconnection to be stable in at least 50% during the overlapping period but accounts for fluctuations. If the correlation sign varies too much (swings between positive correlation to anticorrelation and vice versa) and those changes occur frequent, it won't be considered as a stationary signal as indicated by the binary scores in Table S1. We clarified and added this relevant information to the table caption.*

[p5, 9, and relevant to multiple other places]. Statistical significance is mentioned here for the first time. Why 0.1 (seems a fairly weak test) and how are significance levels adjusted for autocorrelation?

*In this study, the significance level has been chosen fairly weak (p<0.1) for two reasons. First, we are using moving windows and focus on the consistency across successive periods. The number of samples is limited and based on 30-year moving windows. Due to the moving approach, correlations rely on partial overlapping periods, therefore we didn't correct for autocorrelation but focused on the interquartile range. The sign test assesses the temporal stability of the link between climate driver and NRM precipitation, and climate driver and proxy records only and is therefore relatively independent*

of the chosen significance level. Secondly, the sign test and correlation approach has been applied in order to act as a first stage "screening" approach. It limits the pool of available proxy records to be a possible predictor for a particular NRM region. The pre-selection procedure is comparable to a distances screening and probabilities commonly applied by the point-by-point approach (Cook et al., 1999). The final reconstruction and its significance level relies on a bigger sample size (common period 1900-1984) and might be considered more robust.

Critical. [p5, 17–24]. This section describes the reconstruction methodology. The credibility of the work rests on this, so the reader needs to thoroughly understand the details of what has been done. There is not sufficient detail for me to be sure I completely follow what has been done. While it is appropriate to lean on other references for comprehensive treatment (but relevant cited important references are missing from the references) the onus is on the authors to present sufficient details here. The Tierney et al level of detail is a useful model in this context.

[p5, 24]. How spliced?

Yes, same as Tierney, sliced together on the most replicated nest. We added p5 line 24:

*"Nests are spliced together based on the most replicated nest to form a continuous reconstruction."*

[p5, 27–28]. 52, 33 years. At face value 1934–1984 & 1900–1933 gives 51, 34. Missing something?

Yes, we will correct that to p5 27-28:

*"During the common period, 60% of the data are used for calibration (equal to 51 contiguous years) and the remaining 40% (34 years) are used for verification."*

[p5, 30]. ". . .not entirely independent. . ." could be interpreted as mostly independent, which is incorrect.

Agreed and removed from to p5 line 30:

*"These different, but not independent, calibration and verification periods are used to build an ensemble of seven reconstructions for the warm and cool seasons. "*

3.2 Analysis

[p6, 7–9]. Rationale for this analysis? I don't know what normalized trends means in this context.

Agreed, sentence has been completed as:

*"All trends are normalized by the maximum occurrence and presented in histograms."*

[p6, 14–16]. Detail redundant here (provided in Table 1).

We agree, and refer the reader to Table 1c.

[p6, 18]. Deciles need a time interval (e.g. 36 months).

Agreed, we added a statement on p.6 line 22:

*"The individual duration of deciles depends therefore on the time interval covered by each individual drought, as given in Table 1c."*

[p6, 19–21]. Can this be rephrased for clarity? 4.1 Regional climate driver influences

Rephrased as:

*"4.1 Influence of climate drivers on regional rainfall"*

[p7, 13]. ENSO "stands out" only in the warm season. The cool season map is mostly red, but this is misleading when the more nuanced bar graph results are considered. See later comments on Figure 2.

The map shows the highest correlation only and ENSO stands out for both seasons, although weaker in the cool season.

[p7, 14]. 44% < "most", so presumably you mean something else.

We changed the sentence to:

*"Indeed, ENSO explains the greatest proportion of low-latitude rainfall variance during the peak-intensity season (warm season) of up to 44% in the Wet Tropics."*

[p7, 24]. SSWF has the only warm season yellow (IOD) bar.

Yes, that is expected. The Indian Ocean Dipole is important in the June-Oct period in the southwest and southeast. That is consistent with (Risbey et al., 2009) showing that the IOD has its most prominent impact on Australian rainfall during the wet period and located to the south.

4.2.1 Reconstruction skill

[p8, 3, 5]. Figure 3 panel labels are given in the text, but are not shown on the figure.

The figure caption includes the information/labeling abbreviation.

Major. [p8, 5–8]. Some clarification of this earliest year comment is required. First, al- though it doesn't say so in the text, the Figure 3 caption indicates that statistics relate to calibration rather than verification statistics. Wouldn't the latter be better? Second, why is half the maximum calibration variance explained an appropriate metric here, rather than a fixed R2 threshold? Third, given that you can only assess skill based on com- paring with observations, I assume that the early dates relate to how reduced data sets (corresponding to nests) perform against the instrumental data. If this is incorrect then some additional explanation is required. Whether correct or not, have you taken into account degraded proxy performance with time outside of the calibration/verification period? Loss of sample depth, and thus signal, is characteristic of the tree ring data, so there is more to reduced performance than simply the number of proxies. Probably nothing much you can do about this, except to note that the early dates will be inflated (too early), but to an unknown degree.

The reviewer is partially correct; the statistics do not relate only to the calibration statistics but also to the verification statistics. We clarified this in Figure 3 caption by adding respectively to p27 line 9:

*"the year in which R2v and R2c reduce to half of their maximum value, respectively."*

We decided to indicate the degradation of reconstruction performance by a single year for the explained variance only. It should be noted that the given years/numbers, which are based on R2v and R2c, do not serve as a threshold. Those years serve only as indicators how fast/slow the stitched reconstructions of any given region and season half their maximum skill (of course depending on different nests with fewer proxies).

However, the RE and CE years indicate indeed the starting years that has been used as some kind of

threshold for the skillful portion of the reconstruction as stated in p.6 line 4-5.

We clarify this by adding a statement to p. 8, line 9:

*"The CE years indicate the* skillful portion of the reconstruction (CE>0) for further analysis. *"*

4.2.2 Reconstruction time-series

[p8, 22]. Probably best to delete ". . .and past centuries (Fig. 5)", because all comments in this paragraph relate to Figure 4.

Okay

[p8, 29]. Define "low-frequency". I am struck by the lack of it.

We agree and specified the term in p.8 line 29 as followed:

*"Decadal-scale frequency variability (decadal) is evident in all warm and cool season reconstructions."*

[p8, 26–27]. Perhaps I am missing something here, but doesn't your rescaling methodology force this? If so, then this is not a relevant comment.

The CPS reconstruction method scales on the mean and standard deviation which alters amplitudes but not necessarily the upper percentiles (extreme years) correctly.

Major. [p9, 15–27]. This is an interesting approach, but I am unconvinced by the interpretation. Because 30 years is a fairly short window, I suspect that analysis of serially-correlated random numbers may give similar results to what you see here. If so, then the patterns identified cannot realistically be interpreted in the manner done, although the conclusion would be the same. I am not convinced that it amounts to ". . .an additional verification measure".

We totally agree that it is not an additional verification measure and rephrased the sentence to:

*"This indicates a degree of independence between the seasonal reconstructions."*

4.3.1 Contextualising recent rainfall trends

[p10, 7–22]. Apart from the apples vs. oranges caveat (see discussion of Figure 6), this seems OK, but it does beg the question why 30/50 year trends are a key metric, rather than, say, 30/50 year means and variance. See previous comment about providing the rationale for this aspect of the methodology.

We agree with the reviewer that these trends are not a key metric. As far we are concerned 30/50 year running means would indicate the similar results. We refer to trends because our study aims to contextualize often cited recent trends in the literature based on short instrumental records. We do not claim an extensive attribution of recent trends but we use our reconstruction to get an extended estimation of the range of natural variability.

4.3.2 Contextualising the spatial extent and intensity of past droughts

[p10, 31–32]. Surely two droughts are not enough to make such a relatively bold statement, especially since the reconstruction gets the significance of the two droughts around the wrong way (gridded AWAP shows WW2 drought is more significant, but reconstruction indicates the Federation drought).

We agree with the reviewer and added to the sentence:

*"our reconstruction depicts the intensity of the two drought events during the instrumental period quite well."*

Major. [p11, 11–19]. Figure 7 is nice, but here are confusing elements to the results that require explanation. Recon (1900-214) shows central region (R) below average for both the WW2 and Federation droughts. Recon (1600-2014) has WW2 average and Federation very much below average. While I appreciate that deciles are a moving target, drilling down into the results is needed to make sense of what is going on. At face value, Recon (1600-2014) lacks credibility, because it relegates arguably the most significant drought of the instrumental record, based on the instrumental data, to relative insignificance!

We agree with the reviewer and corrected the figure labels. Deciles, its subdivision for plotting reason into 7 categories and a coarse spatial resolution removes some detail. We correct the figure caption of the reconstruction from Recon (1900-2014) to Recon (1900-1990) to make clear that we didn't reconstruct the Millennium drought but sets it into a long-term context.

4.3.4 Extreme years in a long-term context

[p12, 27–28]. This is pushing the envelope, but I am not convinced that you have actually shown that the reconstruction is actually up to this rather demanding task. I would need to see that the instrumental extreme years are captured in roughly the appropriate order.

We agree and tested those claims before. The extreme years during the overlapping period are captured roughly in the same order. Figure 4 indicates some of them.

[p13, 2–16]. Some of this material may be better in the discussion.

We agree and move those lines into the discussion.

[p13, 18–28]. Results in this section have to be taken at face value because the tabled presentation is not well suited to "seeing" the claimed patterns.

We agree with the reviewer and will show each individual year by a map with a supplementary video.

4.6 Comparing our reconstruction with previously published

[p13, 33–34]. Please be explicit about the degree of overlap (%).

We agree and added the degree of overlap to the text:

*"(~60%)"*

[p14, 4]. Is linear correlation against the PDSI appropriate? I don't recall if the PDSI scales linearly and it is also a water balance approach, so has significant memory. My point here is that you might be short changing yourselves by an overly simplistic inter-comparison.

We agree that the correlation analysis might be very simplistic and further analysis is required. We add a statement to p.14 line 5:

*"Figure 9 shows the linear correlations not accounting for memory effects between the austral summer season (DJF) PDSI reconstruction (ANZDA) with our cool and warm season rainfall reconstructions"*

[p14, 8–10]. I don't understand where you are going with this the last sentence. It reads like a criticism of the PDSI, but I suspect that is not your intention. The temperature dependence targets evaporation, making PDSI arguably a superior drought index. And the spatially unresolved parts

presumably relates to the point-based approach, which is also arguably superior (you certainly have not convinced me otherwise).

*Yes, it is not our intention to criticise the PDSI. We have to keep in mind that we are comparing apple and oranges as our reconstruction is calibrated to rainfall not PDSI. It is not expected to align perfectly and highlights aspects of both, the PDSI and rainfall reconstruction. We add this statement to underline those differences to p14 line 2-3.*

*"In addition, the ANZDA drought reconstruction differs substantially in its temporal (DJF) and spatial resolution (gridded), since it is a point-by-point gridded reconstruction, mostly verified for Eastern Australia and targets the PDSI, which accounts not only rainfall but also temperature-depending effects during summer (DJF) and memory effects accounting for soil moisture. The PDSI is therefore more likely to reflect agricultural droughts than meteorological droughts observed in rainfall. "*

Major. [p14, 10]. The poor warm season agreement with the PDSI analysis, except for one region, is quite alarming, especially the near-zero relationships in regions containing the cities where most Australians live. Given that this affects the perceived credibility of Australian drought reconstruction, it might be appropriate to follow up on this here, or in the discussion.

*We partially agree but don't see a strong disagreement with the PDSI reconstruction as we are comparing two different targets, spatial and temporal resolutions. We should highlight again, that the PDSI reconstruction covers mostly Eastern Australia and is highly spatially resolved. Regions containing the cities where most Australians live e.g. Sydney, Melbourne, Canberra, Adelaide are not well represented by the ANZDA reconstruction either. Areas such as the Murray-Darling Basin, which collects and drains off much of its precipitation into catchments for major cities, shows agreement.*

[p14, 14–17]. The cool season SE results are encouraging (water resources implications), but not so the dry season. Coupled with the poor agreement with the drought atlas, and the unconvincing relationship with the coastal records, I'm left doubting the credibility of the reconstruction.

*The drought atlas is based on PDSI and even during the instrumental period, the PDSI differs strongly from rainfall among others due to its temperature component and memory effect. We added a few more comments on the differences including the different seasonality (see above) to remind the reader.*

5 Discussion and conclusion

[p15, 6]. "Eastern Australian" is too broad a phrase – agreement is much more spatially restricted. Personally, I think "high-level" is overselling things. Given that you are reconstructing the same thing (drought) from significantly similar data sets, I was expecting to see most variance in common, and you are well shy of that.

*We disagree with that and suspect confusion about the cool and warm season. We reconstruct rainfall, not drought, which might come closest to a description as meteorological drought. Correlation analysis shows highly significant connections between ANZDA and the rainfall reconstruction during the warm season close to the East Coast and Central Slopes of at least r = 0.6. This becomes clear from figure 9. To avoid further confusion, we increase the font size of the season names in figure 9.*

[p15, 7–8]. I suggest you limit the "compared well" comment to the cool season.

*We disagree with that. See comment above.*

[p15, 11]. Interesting comment about highlighting the quality, because to me they high- lighted the limitations.

A comparison with early documentary records is only possible for Southeastern Australia where we have early records. That is indeed limiting for other regions but provides opportunities for comparison with Southeastern Australia.

[p15, 14]. This is a reasonable statement. But not picked up is some notable evolution in patterns for some regions. For example, MN & R in Figure 6 appear to have increased variability in the late 20th c. Is this real, or a splicing artefact?

That is true and an interesting point for future work. This increase of variability is actually also visible in the instrumental data. Further work is need to investigate this increase in variance. For example, the Rangeland cover the largest region, mostly deserts but not many instrumental observations. Further research could investigate if those changes reflect actual increase in variability or are an artifact of inhomogeneities in the underlying AWAP dataset.

[p15, 14–]. I remain unconvinced by this regression slope analysis approach. It can tell you about the rate of change and its significance, but is that really the important metric in terms of the process explanations you then mention? It also misses important cumulative impacts. For example, the SS and SSWF results show a cumulative decline to a mean substantially lower and with the most extreme droughts all relatively recent. MN and R show the reverse. A different type of analysis would be required to assess the significance of these changes.

We agree with the reviewer and don't claim any assessment of significance or process explanation. We set the most recent trends into a longer term context, which is often not possible with short instrumental records. This approach provides useful information about how usual or unusual those decadal-scale changes are but doesn't refer to processes causing those changes.

Major. [p16, 2–3]. Comparison of instrumental vs. reconstructed trends can only reasonably be made with relevant caveats associated with the pre-processing of the palaeo data. Pre-processing has likely reduced supressed multi-decadal trends, so your histograms in Figure 6 will be pulled in at the tails, which clearly will affect your assessment of how the instrumental data trends (which have not been similarly treated) compare. Note though that recognising this actually reinforces your conclusion about recent trends being within the range of natural variability.

We don't agree with the reviewer on this point. All proxy data has been used as provided by the original author and described in cited publications. We did not pre-process or filter the data other than as done by the original publication by removing non-climatic trends (e.g. biologic growth trend in tree-rings).

Major. [p16, 9–10]. The discussion in this paragraph follows on and emphatically restates earlier comments about the quality of the reconstruction of historical droughts that I think can reasonably be challenged (see [p11, 11–19], above). First, surely you only have two droughts. The millennium drought is outside your proxy data period, so it is essentially spliced instrumental data, is it not? If so, then agreement of spatially- averaged instrumental data with the original gridded data is meaningless, although it does point to issues with spatial units that are too large (a paragraph discussing this spatial scaling issue would be appropriate). For the other two droughts, you can only really claim good agreement for the Federation drought. As previously stated, I think the WW2 drought reconstruction is severely awry, and suggests to me that the methodology may only be suitable for capturing some types of drought (perhaps some additional forcings are not captured by the proxy network). The credibility of the reconstruction is challenged by the poor representation of what the instrumental data shows to be the most extreme drought in the instrumental period (Figure 7, left column).

We agree with the reviewer and add a statement highlighting the spatial scaling issues to the

discussion p.16 line 10:

*"The major droughts during the instrumental period are well represented in terms of their spatial extent, intensity and duration considering the reduced spatial representation of regional averages."*

We can only compare the WW2 and Federations drought during the instrumental period, which is limiting but both droughts are reasonably well captured by comparing the gridded AWAP (first col), and NRM AWAP (second col) and reconstruction (third col). Again especially South Eastern Australia, Central Slopes and East Coast but also the Rangelands agree well. Great differences show up when looking at the WW2 in a long-term context (fourth col) but again Southern, Central and Eastern Slopes show still conditions well below average.

 [p16, 27–33]. This is an interesting point. Can you relate it back to the drivers?

Good point and referred as further work in the discussion as followed:

 *"Independent palaeoclimate reconstructions could help to infer the rainfall amounts back to the climatic drivers causing these diverse drought characteristics and could provide important insights into the climate modes."*

Major. [p17, 3–4]. I don't disagree with this, but it does presuppose that teleconnection patterns will remain stable in a future warming world. The flip side of this is that the reconstructions extend back into a globally cooler period. If teleconnection were different then (and there is evidence to suggest they were for some of your proxies), what then are the implications for your reconstruction (because the transfer function will be wrong. Moreover, drought is not just rainfall. Australian researchers have shown that droughts have intensified in response to increasing T (and thus evaporation), have they not? So a rainfall-only analysis is only part of the story. Surely worth some serious commentary.

We totally agree that droughts are not only a consequence of suppressed rainfall but also temperatures and subsequently evaporation. We added a comment into the discussion part by

*"It should be pointed out that our seasonal reconstruction represents rainfall only. Droughts are a result of complex interactions of various atmospheric variables and interactions of different time-scales. Important factors contributing to droughts are temperature, soil moisture and evaporation, which are not accounted for in this reconstruction."*

Table 1. SAM is missing. Additional details of indices would be useful (e.g. I assume NCT and MWP are SST based). A sentence or two describing each index would be useful. Surprising you have not included a west pole pressure index.

Yes, we will add SAM and its reference. For details on the individual indices we refer the reader to the original publications. The information about pressure conditions over the west pole is included in the calculation of the SOI due to pressure differences between Darwin and Tahiti and therefore not further being looked at in this study.

Table 2. The caption could usefully be reworded for clarity. Is the information for "Instru" the reconstructed data for the instrumental period, the same but with instrumental data spliced on the end, or the instrumental data? This seems a rather ineffectual way of presenting the information – visualisation would highlight temporal patterns, temporal clustering, and inter-regional patterns in a way that tabled numbers do not.

We agree, and will reword the caption.

Figure 1. See previous comment about inability to resolve the proxies and the regions on this map.

Also, given that many proxies were eliminated, would it not be more useful to limit the map to those proxies that actually end up being used. Moreover, it would be interesting to see this broken down by region in the supplement. Without a laborious process of extracting the relevant information from Table S1 and remapping, this useful information about contributing proxies is unavailable to the reader.

We agree, and will add a supplementary figure showing all contributing proxies on a map

Figure 2. It appears that the SOI is generally superior or comparable to the other three ENSO indices. That point could be made in the text and this figure simplified. There is a wealth of information in the bar charts, but the maps are unsophisticated in the treatment of this, and I think counterproductive in oversimplifying matters. I don't recall comment in the text related to the logic of pooling SAM and BLK. It would be useful to include the region codes along with their long names on the bar charts (also on the maps). Consider adding a horizontal line separating the two parts of the figure. Because you don't have axis labels, you need to explicitly state in the caption that the bar graphs show correlations. See previous comment about uncertainty about whether autocorrelation has been allowed for in the significance levels cited (see [5, 9] above).

We agree that the SOI is generally superior. Great point! Maybe its direct computation from atmospheric pressure shows stronger connection with atmospheric processes causing rainfall anomalies in Australia. We will add a horizontal line separating the two parts of the figure, add the code names for the regions and add a statement about the correlations. The pooling was applied only for visualization purpose to draw distinctions amongst tropical (ENSO and IOD), subtropical (STRL and STRI) and extratropical drivers (SAM and BLK). We will add this information to the caption.

Figure 3. It would be useful to have the years corresponding to the plotted statistics shown.

We refer to some of the corresponding years already on page 8, line 10-20.

Figure 4. There are several instances where the reconstruction is outside the ensemble range. Having gone to the trouble of calculating the ensembles (a good thing), why isn't a mean/median (or other measure of central tendency) used for the reconstruction? Doing so would "fix" some of the points of difference with the instrumental data (e.g. in MB, MN, WT). It would introduce other issues, but the net benefit may be positive, and a transfer function based on the full data rage may be more robust. Just a thought.

Thanks for this comment and finding this error! We mislabeled the instrumental data with the reconstructed data in figure 4 and will change that plot. The reconstruction falls within the ensemble range, so sorry for this. We haven't looked at the ensemble mean/median caused we would have to deal with other issues such as an inadequate representation of amplitudes due to the averaging process.

Figure 7. I presume that the millennium drought is "missing" for Recon (1900–2014), because you only have instrumental data. If that is true, I don't get why it appears in Recon (1600–2014) – its not reconstructed, its spliced instrumental data isn't it? Need to specify the time periods for decile calculation (12/24/36 months?).

Deciles are computed according to the duration of the drought we are looking at. The duration of droughts is indicated by their start and end years in table 1c.

Figure 8. Please expand the caption to better explain exactly what is being shown here. How is the starting point for each drought determined?

We agree and added to the caption:

*"Decile Plots for a) significant drought periods (according to table 1c)…"*

Minor points

We will account for all minor points.

[p2, 8]. Delete "a" at the end of the line.

[p2, 10]. This style of referencing with a list of references at the end of a paragraph is unfortunate. They presumably don't all relate to the last point, and if they do then there are missing references in the body of the paragraph.

[p3, 27]. Do you mean "Compile"?

[p10, 32]. New paragraph (millennium drought)?

[p11, 11]. [somewhat] similar?

[p11, 33]. provide[s] insight.

[p12, 3]. Suggest you change "many" to "several"? 4–5/8 and only cool season.

[p12, 14]. Seems to [be] a result.

[p12, 24]. Breaking up paragraph into smaller ones would help readability.

[p12, 26]. Expand "Black Thursday" for benefit of non-Australian readers.

[p14, 5]. Reconstruction[s]?

[Table 2, 4]. referred [to] as.

[Figure 8, 3]. Delete "a" (3rd last word).

[Figure 9]. Fenby et al should be Fenby and Gergis.

Cook, E. R., Meko, D. M., Stahle, D. W. and Cleaveland, M. K.: Drought reconstructions for the continental United States*, J. Climate, 12(4), 1145–1162, 1999.

Fiddes, S. and Timbal, B.: Assessment and reconstruction of catchment streamflow trends and variability in response to rainfall across Victoria, Australia, Clim. Res., 67(1), 43–60, doi:10.3354/cr01355, 2016.

HUTCHINSON, M. F.: Interpolating mean rainfall using thin plate smoothing splines, International journal of geographical information systems, 9(4), 385–403, doi:10.1080/02693799508902045, 1995.

King, A. D., Donat, M. G., Alexander, L. V. and Karoly, D. J.: The ENSO-Australian rainfall teleconnection in reanalysis and CMIP5, Climate Dynamics, 44(9-10), 2623–2635, doi:10.1007/s00382-014-2159-8, 2014.

KOCH, S. E., DESJARDINS, M. and KOCIN, P. J.: An Interactive Barnes Objective Map Analysis Scheme

for Use with Satellite and Conventional Data, Journal of Climate and Applied Meteorology, 22(9), 1487–1503, 1983.

Lewis, S. C. and LeGrande, A. N.: Stability of ENSO and its tropical Pacific teleconnections over the Last Millennium, Clim. Past, 11(10), 1347–1360, doi:10.5194/cp-11-1347-2015, 2015.

Pook, M. and Gibson, T.: Atmospheric blocking and storm tracks during SOP-1 of the FROST Project, Australian Meteorological Magazine, 51–60, 1999.

Risbey, J. S., Pook, M. J. and McIntosh, P. C.: On the remote drivers of rainfall variability in Australia, Monthly Weather …, 137(10), 3233–3253, doi:10.1175/2009MWR2861.1, 2009.

Timbal, B. and Drosdowsky, W.: The relationship between the decline of Southeastern Australian rainfall and the strengthening of the subtropical ridge, Int. J. Climatol., 33(4), 1021–1034, doi:10.1002/joc.3492, 2012.

Timbal, B., Arblaster, J. M. and Power, S.: Attribution of the late-twentieth-century rainfall decline in southwest Australia, J. Climate, 19(10), 2046–2062, 2006.

---

## Author Comment (AC3)

Essential additions for this paper:

(1) Add a "Data Availability" section to include a URL/Data Citation to a landing page that lists the datasets used in this paper (Table S1) and the URL/Data Citation for the primary output of this study.

We agree and will add the data availability information to the landing page. Does Climate of the past have an opportunity to provide a URL/Data Citation for the primary output internally?

(2) Add Data Citations or URLs (in addition to publication citations) for each of the 185 records used in for the rainfall reconstructions in this study. For those records not already in a persistent public repository, submit the essential metadata along with the proxy data and add the corresponding Data Citation (or URL) in Table S1.

We agree and will add all data citations and metadata to an additional Table in the supplement. The 185 chronologies used in our analyses are the products of several different research groups and are in various states of pre- and post-publication Most of the records have already been lodged in publicly available repositories (typically the ITRDB). In a supplementary file, we provide the essential metadata for the remaining chronologies. These include: chronology name, site/location, length, type, and collector/contact person.

(3) Submit the primary outcome of this study, the regional rainfall reconstructions for cool and warm seasons (Fig 5), to a public repository and include the Data Citation in "Data Availability".

We will do that

(4) Archive the instrumental time-series targets for the reconstructions (Fig 4) along with the reconstructions.

We will do that

---

## Author Response (AR2)

Dear Hans,                                                                17.10.2017

We have made the requested changes which included:

1. Re-formation of reference list
    1.1. Removal capital letters: Koch and Hutchinson
    1.2. Included the Melvin et al 2008 as a reference for the signal free standardization
    1.3. Removed redundant citation Ummenhofer et al 2015b
2. Update of Supplementary table
    2.1. Replaced all N/A with URL or doi
3. Inclusion of data availability section
    3.1. Submitted primary input and output are archived in a permanent archive

The primary input and output data will be available on figshare following this link:
https://figshare.com/s/a73ff374933b07c7e13c, which ensures a permanent data citation with :
doi: 10.4225/49/59e3ee30cdbbc

Sincerely,

Mandy Freund,
on behalf of the authors